# communications
# earth & environment

# Anthropogenic sulfate aerosol pollution in South and East Asia induces increased summer precipitation over arid Central Asia

Xiaoning Xie [1,2 ✉], Gunnar Myhre [3], Drew Shindell [4], Gregory Faluvegi [5,6], Toshihiko Takemura [7], Apostolos Voulgarakis[8,9], Zhengguo Shi[1], Xinzhou Li [1], Xiaoxun Xie[1], Heng Liu[1,10], Xiaodong Liu [1,11] & Yangang Liu [12 ✉]

Precipitation has increased across the arid Central Asia region over recent decades. However, the underlying mechanisms of this trend are poorly understood. Here, we analyze multi-model simulations from the Precipitation Driver and Response Model Intercomparison Project (PDRMIP) to investigate potential drivers of the observed precipitation trend. We find that anthropogenic sulfate aerosols over remote polluted regions in South and East Asia lead to increased summer precipitation, especially convective and extreme precipitation, in arid Central Asia. Elevated concentrations of sulfate aerosols over remote polluted Asia cause an equatorward shift of the Asian Westerly Jet Stream through a fast response to cooling of the local atmosphere at mid-latitudes. This shift favours moisture supply from low-latitudes and moisture flux convergence over arid Central Asia, which is confirmed by a moisture budget analysis. High levels of absorbing black carbon lead to opposing changes in the Asian Westerly Jet Stream and reduced local precipitation, which can mask the impact of sulfate aerosols. This teleconnection between arid Central Asia precipitation and anthropogenic aerosols in remote Asian polluted regions highlights long-range impacts of anthropogenic aerosols on atmospheric circulations and the hydrological cycle.

[1] SKLLQG, Institute of Earth Environment, Chinese Academy of Sciences, Xi'an, China. [2] CAS Center for Excellence in Quaternary Science and Global Change, Xi'an, China. [3] Center for International Climate and Environmental Research, Oslo, Norway. [4] Nicholas School of the Environment, Duke University, Durham, USA. [5] Center for Climate System Research, Columbia University, New York, NY, USA. [6] NASA Goddard Institute for Space Studies, New York, NY, USA. [7] Climate Change Science Section, Kyushu University, Fukuoka, Japan. [8] Department of Physics, Imperial College London, South Kensington Campus, London, UK. [9] School of Environmental Engineering, Technical University of Crete, Chania, Crete, Greece. [10] Xi'an Institute for Innovative Earth Environment Research, Xi'an, China. [11] College of Earth and Planetary Sciences, University of Chinese Academy of Sciences, Beijing, China. [12] Environmental and Climate Sciences Department, Brookhaven National Laboratory, Upton, NY, USA. ✉email: xnxie@ieecas.cn; lyg@bnl.gov

Arid Central Asia (ACA hereafter) is located in the inner-most central part of the Eurasia continent, and is composed of Central Asian countries and Northwest China, where there exist several deserts with a total annual precipitation less than 200 mm (e.g., Taklamakan and Gobi deserts). Observational evidence has revealed a pronounced increase of precipitation over ACA since 1961, occurring mostly in summer and winter[1–4] (also see Fig. 1a, b). These observational results about the ACA significant wetting challenge the paradigm of "dry gets drier, wet gets wetter" under global warming[5–9].

The ACA wetting during recent decades has been attributed to external and internal drivers in previous studies. One mechanism proposes that this regional wetting is caused by global warming associated with increased greenhouse gases (GHGs) through enhancing convective precipitation or extreme precipitation[3,10–12]. Another mechanism proposed is related to regional accelerated snow and glacier melting in high-mountain Asia that has been supplying more moisture into the atmosphere over ACA during recent decades[1,13]. Internal natural variability has been proposed to explain this decadal change of precipitation including inter-decadal variations of sea surface temperature and atmospheric large-scale circulation[14,15]. However, these drivers of ACA regional wetting remain highly controversial, especially in summer, and cannot explain the southern shift of Asian Westerly Jet Stream (AWJS) (Fig. 1c), which is closely associated with the ACA summer wetting[4,16,17]. As shown in Fig. 1d, increasing GHGs by doubling $CO_2$ only result in the winter and spring precipitation increase over ACA, but do not lead to a significant AWJS equatorward shift or to a regional precipitation increase in summer[18], based on the multi-model results from the Precipitation Driver and Response Model Intercomparison Project (PDRMIP, see "Methods").

Recent studies have indicated that anthropogenic aerosols can induce global and regional changes in precipitation through aerosol-radiation and aerosol-cloud feedbacks[19–23]. Anthropogenic aerosol emissions have exponentially increased due to the rapid industrialization and economic development over Asian polluted regions including the Indian subcontinent and eastern China during recent decades[24–26] (also see Supplementary Fig. 1 about the trend of aerosol optical depth (AOD)). These polluted regions are mostly located in the South and East Asian monsoon regions. These anthropogenic aerosols significantly affect the South and East Asian summer monsoon precipitation through changing the land-sea thermal gradients and cloud microphysical processes[27–36]. However, the effects of remote anthropogenic aerosols on the regional precipitation have been ignored over ACA in previous studies, probably because the ACA is far away from these polluted regions. Here, we show that anthropogenic aerosols over the remote Asian polluted regions significantly enhance the ACA summer precipitation through driving an equatorward shift of AWJS, by analyzing the sensitivity experiments of a tenfold increase in present-day sulfate concentrations across the Asia region (60°E–140°E, 10°N–50°N) in PDRMIP (experiment denoted as SULx10Asia). Furthermore, the results are compared with the simulations of a tenfold increase in black carbon aerosols over the Asia region (BCx10Asia) and doubled $CO_2$ (CO2x2).

## Results

**Asian sulfate-induced precipitation change**. Figure 2 shows the Asian sulfate-induced changes in the summer precipitation based on multi-model experiments of PDRMIP. Consistent with previous studies with global climate models (GCMs)[30–36], Asian sulfate aerosols significantly weaken the South and East Asian summer

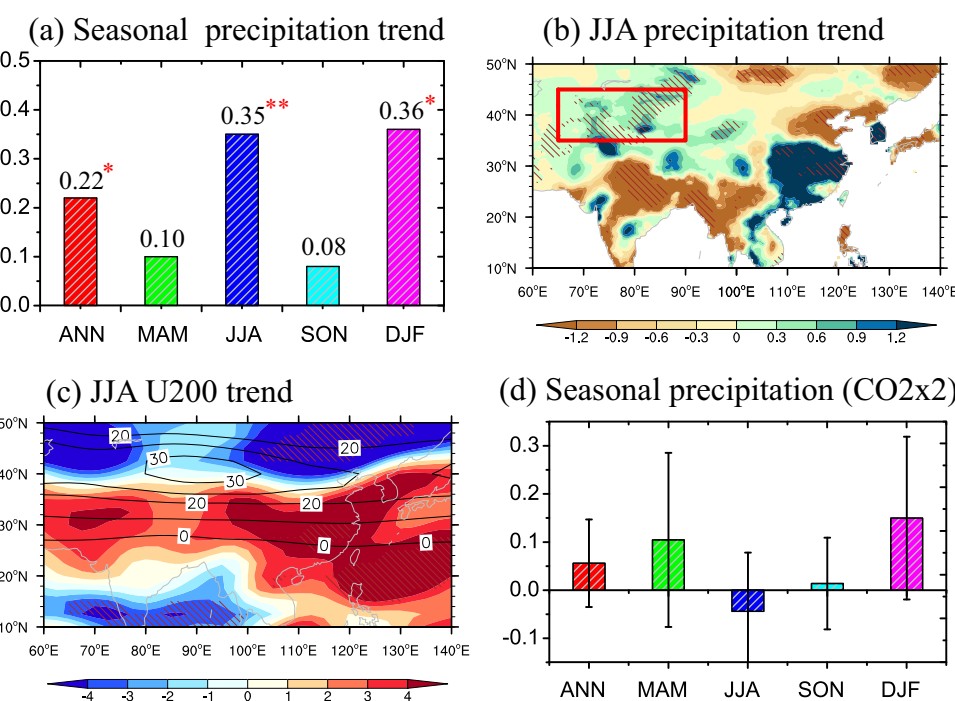

**Fig. 1 Precipitation and westerly wind (U200) trends in recent decades over Arid Central Asia (ACA). a** Seasonal precipitation trends (mm day⁻¹ per 100 years) over ACA in annual (ANN), March-May (MAM), June-August (JJA), September-November (SON), December-February (DJF) from 1961 to 2005 based on CRU TS V4.04. Spatial pattern in trends of (**b**) JJA precipitation and (**c**) JJA westerlies at 200 hPa (U200, m s⁻¹ per 100 years) during the same period (NCEP/NCAR Reanalysis 1). **d** Seasonal changes induced by GHGs (CO2x2) in precipitation (mm day⁻¹) for the multi-model mean (MMM) in PDRMIP. Red box in (**b**) indicates the ACA region and thin black lines in (**c**) show the climatological westerlies. The single and double red stars in (**a**) represent significance at the 95% and 99% confidence level by a standard *t* test, respectively. Slanted lines in (**b, c**) represent significance at the 95% confidence level. Error bars of MMM in (**d**) represent the standard deviation among the PDRMIP models.

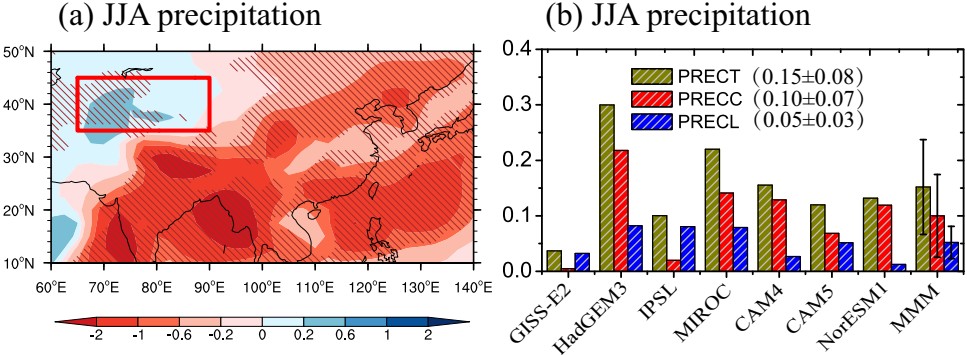

**Fig. 2 Asian sulfate-induced changes in precipitation over arid Central Asia (ACA). a** Geographical distribution of changes induced by increasing Asian sulfate (SULx10Asia) in JJA precipitation (mm day$^{-1}$) in the multi-model mean (MMM), and (**b**) for changes in JJA total precipitation (PRECT, mm day$^{-1}$), convective precipitation (PRECC, mm day$^{-1}$), and large-scale precipitation (PRECL, mm day$^{-1}$) averaged for the ACA domain. Slanted lines in (**a**) indicate where the MMM is more than 1 standard deviation away from 0 and error bars of MMM in (**b**) represent the standard deviation among the PDRMIP models.

monsoons and decrease accordingly the summer precipitation over the Indian subcontinent and northern China (Fig. 2a), through reducing the land-sea thermal gradients and changing the cloud-rain autoconversion processes. The results also clearly show a significant increase in ACA summer precipitation (Fig. 2a), despite the fact that the ACA region is far away from the Asian polluted regions (Supplementary Fig. 1). This significant increase in ACA summer precipitation had not been identified in previous work[30–36], due likely to substantial underestimations of anthropogenic aerosol burden and the corresponding AOD over Asia in those GCM studies[37–40]. All the seven individual models indicate a summer precipitation increase over the ACA region in Supplementary Fig. 2. The region-averaged changes are also shown in Fig. 2b, indicating that the total precipitation over ACA is increased by 0.15 ± 0.08 mm day$^{-1}$ (14.3%±6.0% in Supplementary Table 1) in the multi-model mean, which is mainly contributed by the convective precipitation with 0.10 ± 0.07 mm day$^{-1}$ (larger than 65% of total precipitation). Accompanying the increase in convective precipitation, there is a sizeable enhancement of 7.4% also shown in the extreme precipitation (see Methods), mainly attributed to an increase in the frequencies of extreme precipitation events (9.2%) in Supplementary Table 1. These GCM results support the observational findings that the precipitation increase over ACA is attributed mainly to enhancements of convective precipitation or extreme precipitation in boreal summer[3,10–12].

**Asian sulfate-induced AWJS change**. The summer precipitation increase over ACA mainly results from changes of the large-scale circulation (e.g., westerlies) induced by Asian sulfate aerosol forcing. Figure 3 shows the annual and seasonal mean changes in the AWJS position (defined as the meridional position of maximum speed of 200 hPa westerlies between 60°E and 140°E), where the positive (negative) values represent a northward (or southward) shift of AWJS, respectively. Asian sulfate aerosols significantly shift the AWJS to the Equator during all the seasons, especially in summer by almost −2 degrees in the multi-model mean, as shown in Fig. 3a, b. Figure 3c, d shows the spatial distribution and zonal mean of westerlies, indicating a significant increase in westerlies over the regions between 10°N to 30°N and a decrease at higher latitudes (i.e., a southward shift of AWJS). As shown in Fig. 3b, all the PDRMIP models show a southward shift of summer AWJS, except NCAR-CESM1-CAM5 (see also Supplementary Fig. 3). Note that the insignificant shift in summer AWJS in NCAR-CESM1-CAM5 in Supplementary Fig. 3f is mainly owing to complex nonlinear interactions between aerosols and cloud microphysical properties[41–43].

The meridional change of AWJS position induces a low-level circulation anomaly that directly influences the regional surface precipitation. The AWJS southward shift induces an anomalous southwesterly wind at lower levels and an ascending motion prevailing over ACA, according to the PDRMIP multi-model results in Supplementary Fig. 4a, b. These changes in the low-level circulation induced by Asian sulfate aerosols are much more favorable for the water vapor supply from low latitudes around the Indian Ocean and water vapor flux convergence over this region (Supplementary Fig. 4c), compared to the BASE experiments in the PDRMIP simulations. The relationships between the AWJS position and low-level circulation were also evident in interannual scales and interdecadal scales. Based on recent observational studies, at the interannual timescale, there exists anomalous southerly winds and much more summer precipitation due to enhanced water vapor transport at the years with further south position of AWJS, compared with north position[44–46]. Additionally, the interdecadal variability of the AWJS shows a significant southward shift from the Reanalysis (shown in Fig. 1c) and also induces low-level southerly winds (Supplementary Fig. 5), which is closely related to the summer wetting trend during recent decades over ACA[4,16,17]. Therefore, the AWJS-induced changes in low-level circulation result in more summer convective precipitation and also more extreme precipitation over the ACA region, as shown in Fig. 2. Another hypothesis is the "monsoon-desert" mechanism, claiming that the weakened ascent over the Asian monsoon region was directly responsible for the weakened descent over the ACA region[47,48]. This mechanism can also explain the ACA wetting induced by Asian anthropogenic aerosols due to the weakening Asian monsoon, which is also suitable to the CESM-CAM5 result.

To confirm the water vapor source, we further analyze the atmospheric moisture budget along the ACA region[49,50] (see Methods). Quantitative estimates of water vapor flux changes induced by Asian sulfate aerosols across four boundaries over the ACA region are presented in Supplementary Fig. 6. The sign of the moisture budget changes is positive (negative) if the water vapor transports into (out of) the ACA box. Over this arid region, the increases in the water vapor fluxes are mainly from the southern side (0.14 mm day$^{-1}$) and the western side (0.10 mm day$^{-1}$), which confirmed supplying the moisture over ACA from low latitudes through the anomalous low-level southwesterly winds induced by the AWJS southward shift.

The meridional shift of summer AWJS results from the radiative forcing and temperature feedbacks of anthropogenic aerosols[51–53]. Supplementary Fig. 7a shows a significant increase in AOD over

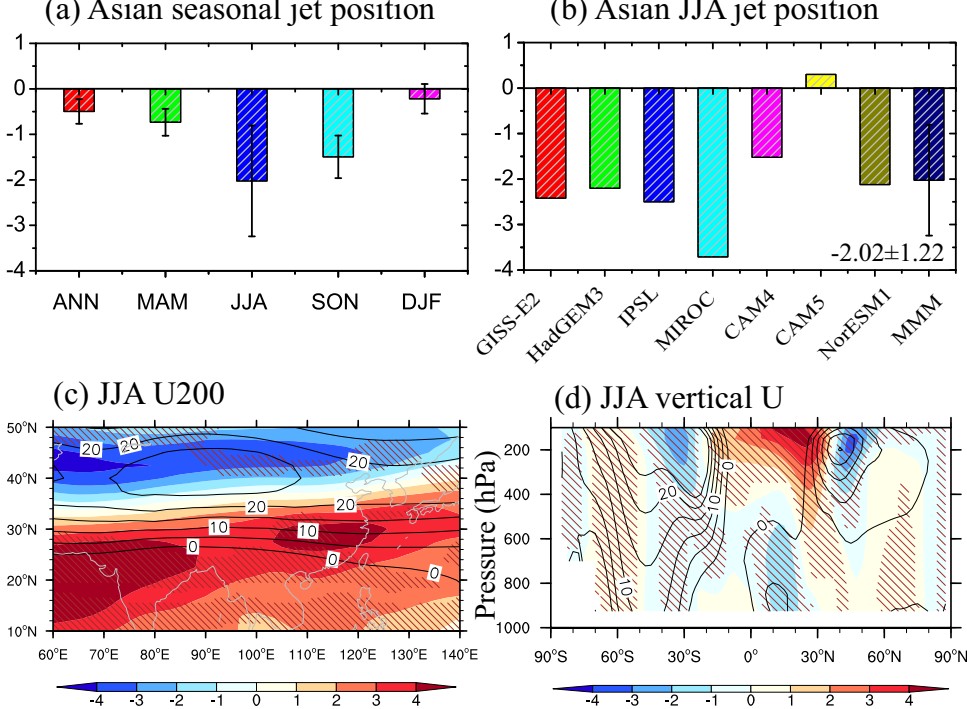

**Fig. 3 Asian sulfate-induced changes in Asian westerlies and westerly jet stream position. a** Changes induced by increasing Asian sulfate (SULx10Asia) in seasonal westerly jet position (degree latitude) for the multi-model mean (MMM). **b** Changes in JJA westerly jet position for individual PDRMIP models and MMM. **c** Spatial pattern of changes in JJA westerlies at 200 hPa (U200, m s⁻¹) for MMM and (**d**) in zonal mean JJA westerlies (U, m s⁻¹). Thin black contour lines in (**c**, **d**) show the climatological westerlies. Error bars of MMM in (**a**, **b**) represent the standard deviation among the PDRMIP models and slanted lines in (**c**, **d**) represent where MMM is more than 1 standard deviation away from 0.

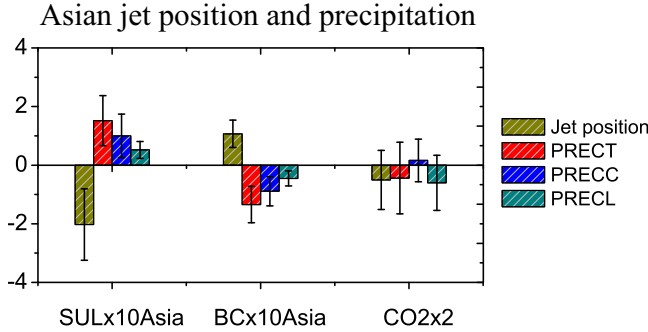

**Fig. 4 Changes in Asian westerly jet stream position and precipitation over the arid Central Asia (ACA) induced by individual climate forcers.** Asian sulfate (SULx10Asia) and black carbon (BCx10Asia), as well as GHGs (CO2x2) induced changes for the multi-model mean (MMM) in JJA westerly jet position (degree), JJA total precipitation (PRECT, $10^{-1}$ mm day⁻¹), convective precipitation (PRECC, $10^{-1}$ mm day⁻¹), and large-scale precipitation (PRECL, $10^{-1}$ mm day⁻¹) over ACA. Error bars of MMM represent the standard deviation among the PDRMIP models.

Asia, especially over the Indian subcontinent and eastern China in multi-model mean (much larger than 0.9 over these two polluted regions). Since anthropogenic sulfate aerosols mainly scatter short-wave radiation, the increase in anthropogenic AOD leads to a significant and large negative effective radiative forcing (ERF) over these regions in Supplementary Fig. 7b. The negative ERF of sulfate aerosols leads to a significant decrease in air temperature over the mid-latitudes, especially in 600-200 hPa levels (Supplementary Fig. 7c), which then yields changes of the meridional temperature gradients, with an increased temperature gradient in mid-latitudes and a decreased temperature gradient in higher latitudes in Supplementary Fig. 7d. These changes in the meridional

temperature gradients drive a significant southward shift of summer AWJS shown in Fig. 3. Additionally, the changes in wind fields over ACA are also consistent with the decrease in 200 hPa geopotential height induced by the local atmospheric cooling (see Supplementary Fig. 8). It is noted that, in NCAR-CESM1-CAM5, there exists larger ERF over the Asian polluted land regions due to a strong direct ERF, but also even larger ERF over adjacent oceanic regions, at lower and higher latitudes (Supplementary Fig. 9), due to indirect ERF from aerosol-cloud interactions. This model is emission-driven, whereas the fixed aerosol concentrations are used in most PDRMIP models. The aerosol emissions from the Asia region will affect the downwind area, which leads to a larger indirect ERF over the adjacent oceanic regions and the insignificant changes in meridional temperature gradients (Supplementary Fig. 10). It should be mentioned that this model significantly overestimates the indirect ERF due to the physical parameterizations of aerosol-cloud interactions e.g., diagnostic precipitation[54] and dispersion effect on cloud droplet effective radius and cloud-rain autoconversion process[43,55], which may artificially amplify the aerosol influence over the adjacent oceanic regions.

**Relative contributions of fast and slow responses.** As suggested by recent studies[56–58], fast and slow responses should be evaluated separately in multi-model intercomparisons, as this helps investigate and understand the causes of responses in the global/regional circulation and water cycle. Therefore, we also examine the relative contributions of fast and slow responses (see Methods) of AWJS and summer precipitation, as shown in Supplementary Fig. 11. It is shown that the fast responses due to rapid adjustments dominate the changes in the AWJS position as well as the total, convective, and large-scale precipitation, whereas the slow responses due to sea surface temperature changes are negligible. That is because those Asian anthropogenic aerosols are

mainly concentrated over land in Asia, which can influence the atmospheric thermal gradient, westerlies, and, in turn, regional precipitation through rapid adjustments due to the local atmospheric cooling. It is noted that sulfate aerosol reduces the surface heat fluxes, in turn leads to the local atmospheric cooling through decreasing the land surface temperature, whereas absorbing black carbon (BC) aerosol warms the local atmosphere by directly absorbing the short-wave radiation.

**Comparisons between sulfate aerosols and other forcings.** Figure 4 compares the climatic feedbacks over ACA induced by sulfate aerosols and by BC and GHGs. Compared with Asian sulfate aerosol, Asian BC exerts an opposite effect on the AWJS position and precipitation through enhancing local atmospheric heating at middle and high attitudes, thereby a northward shift of AWJS and less summer precipitation (see Supplementary Fig. 12a, c). Note that BC yields a relatively smaller change in AWJS position and precipitation in PDRMIP results, due likely to its much lower burden relative to sulfate aerosol[25,59,60]. It is noteworthy that the climatic influence is also dependent on different optical parameters and vertical profiles of aerosols[61,62]. Here, we can separate the impacts of BC and sulfate and show their opposite effect on ACA climate in PDRMIP. However, due to a gap between the idealized 10-fold increase and observational change in aerosols, the PDRMIP cannot quantify the relative contribution of sulfate and BC aerosols in the real world. Additionally, human-induced GHGs have been recognized as an important driver of recent climate change and an important mechanism of ACA wetting during recent decades[3,10–12]. However, as shown in Fig. 1d, increasing $CO_2$ appears to only result in the increase of winter and spring precipitation over ACA, and cannot account for summer precipitation increases revealed in PDRMIP experiments. In terms of spatial distribution in Supplementary Fig. 12b, d, the GHG-induced changes in AWJS and precipitation during summer are small and insignificant. Therefore, these comparisons reinforce the non-negligible role of increasing sulfate aerosols in the remote polluted regions in determining the increase in ACA summer precipitation.

## Conclusions

Significant increases in precipitation over the arid Central Asia (ACA) region has been observed during recent decades, which challenges the paradigm of "dry gets drier, wet gets wetter" under global warming. However, the drivers of the recent ACA wetting remain highly controversial. Through the analysis of the multi-model dataset in PDRMIP, we find that anthropogenic sulfate aerosol over the remote Asian polluted regions including the Indian subcontinent and eastern China significantly enhances the summer precipitation over ACA, and likely contributes to the recent ACA wetting. The main results are schematically summarized in Fig. 5. Asian sulfate significantly drives the summer Asian westerly jet stream (AWJS) to shift equatorward through fast atmospheric responses due to local atmospheric cooling, which is much more favorable for the moisture supply from the ocean and moisture flux convergence over the ACA region. The moisture supply over ACA from low latitudes around the Indian Ocean through anomalous southwesterly winds is further affirmed by the moisture budget analysis. In contrast to sulfate aerosol, absorbing BC aerosol induces a decrease in ACA precipitation and an AWJS poleward shift, which can mast the climatic effect of sulfate. This teleconnection between the ACA precipitation changes and remote anthropogenic aerosols highlights long-range impacts of anthropogenic aerosols on atmospheric circulations and the hydrological cycle that have been overlooked.

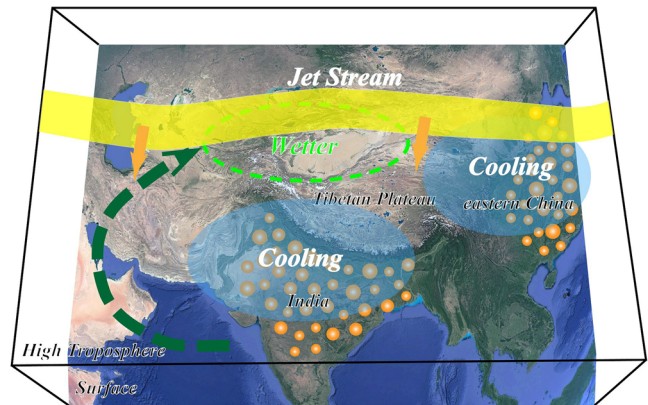

**Fig. 5 Schematic illustration of teleconnection-like effect of anthropogenic aerosols in heavily polluted regions on summer precipitation increase over Arid Central Asia.** The yellow belt represents the Asian Westerly Jet Stream (AWJS), and the corresponding arrows indicate the southward shift of AWJS. The dashed green arrow represents the moisture fluxes from low latitudes through anomalous low-level southwesterly winds. Base map from Google Earth, Landsat/Copernicus (Data: SIO, NOAA, U.S. Navy, NGA, GEBCO, IBCAO, USGS).

Although the idealized PDRMIP simulations make a specific attribution difficult, our analyses suggest that sulfate aerosol can enhance the ACA summer precipitation and it may be an important driver of recent ACA wetting. It is noted that historical GCM simulations significantly underestimate the ACA wetting in the Coupled Model Intercomparison Project Phase 5 (CMIP5) and do not significantly reproduce the southward shift of AWJS during recent decades[32,63–65]. Previous studies show that, mainly due to inaccurate regional emission inventories and aerosol process parameterizations, there are significant low biases of anthropogenic aerosols over Asian polluted regions (especially over the Indian subcontinent and northern China) in GCMs[37–40], which may lead to this underestimation. Based on the latest CMIP6 models with notable improvements in aerosol simulations, recent work shows that anthropogenic aerosols were a potential cause of the changing subtropical westerly jet over recent decades[66] and also implies the regional precipitation increase over ACA[66]. These CMIP6 results support the proposed mechanism of aerosol-induced increase in ACA summer precipitation through affecting AWJS. Additionally, the sulfate and BC aerosols have the opposite effects on the regional precipitation over ACA, which is also evident over the West Coast of the United States[67]. Combined with influence of a large internal variability of ACA precipitation associated with the El Niño Southern Oscillation and North Atlantic Oscillation[14,15], the tug of war effect between sulfate and BC aerosols makes the detection of specific aerosol climatic effect in observations difficult over this region.

Furthermore, regional emissions over Asia have been rapidly changing since 2010, showing a clear aerosol dipole of changes with a significant reduction over eastern China and a concurrent increase over the Indian subcontinent[68–70]. How this change of aerosol emission distributions will affect the ACA precipitation merits further investigation. Additionally, the emissions of anthropogenic aerosols decreased dramatically during the COVID-19 pandemic in the years 2020 and 2021, which may induce the enhanced Asian summer monsoon in these two years[71,72]. We further check the summer precipitation change over the ACA region during these two years. Supplementary Fig. 13 shows an observed negative anomaly of summer precipitation in the years of 2020 and 2021 over almost the ACA region, which is consistent with our results.

## Methods

**PDRMIP data**. We used the numerical experiments of the climate perturbations performed in PDRMIP[23,73] with a doubling of CO2 concentrations (CO2x2), a tenfold increase in Asian sulfate concentrations or emissions (labeled by SULx10Asia), and Asian black carbon concentrations or emissions (denoted by BCx10Asia). Nine GCMs with one baseline experiment (BASE) and the perturbed experiments are CanESM2, GISS-E2, HadGEM2, HadGEM3-GA4, IPSL-CM5A, MIROC-SPRINTARS, NCAR-CESM1-CAM4, NCAR-CESM1-CAM5 and NorESM1[23]. For perturbed experiments with regional aerosols, all the GCMs except CanESM2 and HadGEM2 were performed in PDRMIP. In the PDRMIP experiments, a pair of numerical simulations was conducted for the fixed sea surface temperature (fSST; run for 15 years) and the coupled climate simulations (Coupled; run for 100 years). All the simulated data had the monthly temporal resolution and was interpolated to the horizontal resolution by bilinear interpolation with 2.5°x2.5°, and only the last 10 years of fSST experiments and the last 50 years of Coupled experiments are used to analyze. Note that the effective radiative forcing (ERF) is defined as the total radiative flux differences at the top of atmosphere (TOA) between the BASE and perturbed fSST experiments, which permits rapid adjustments such as atmospheric cooling or heating[74]. Direct ERF is simply calculated as the clear-sky radiative flux differences at TOA between these two fSST experiments, whereas indirect ERF is the cloud forcing differences at TOA.

**Extreme precipitation**. For a given station or model grid point, an extreme precipitation event occurred when the daily precipitation amount was beyond the 95th percentile value of all rainy days in the reference period[75], in which daily precipitation is larger than 0.1 mm[76]. The extreme precipitation (R95p) was calculated as the total precipitation amount during all the events beyond the fixed threshold obtained for the reference period. The changes in R95p can be decomposed into those of frequency and intensity[77]. The change in frequency is represented as the change in precipitation event number beyond the fixed threshold. The intensity change is calculated as the difference between the total precipitation amount in all the extreme events in the reference period and that of the strongest equal number of events in the latest period[77].

**Moisture budget analysis**. According to the methods[49,50], the column integrated atmospheric moisture fluxes $Q_u$ (zonal flux) and $Q_v$ (meridional flux) over the region are diagnosed by using Eq. (1) during the model integration as

$$Q_u = 1/\rho_w g \int_{P_s}^{P_t} \langle qu \rangle \mathrm{d}p$$
$$Q_v = 1/\rho_w g \int_{P_s}^{P_t} \langle qv \rangle \mathrm{d}p \qquad (1)$$

where $\langle \rangle$ represents the monthly mean. The winds $u$ and $v$ are the zonal wind and meridional wind, respectively. $\rho_w$ is the water density and $g$ is the gravity acceleration. $\rho_s$ is the surface pressure, $\rho_t$ is given the value of 300 hPa[78]. The integrated moisture fluxes $Q_u$ (or $Q_v$) are expressed in unit of m$^2$ s$^{-1}$. The column integrated atmospheric moisture fluxes across each domain edge are averaged, and divided by the length along the flux trajectory, where the unit is in mm day$^{-1}$. Here, due to missing the variables of $\langle qu \rangle$ or $\langle qv \rangle$ in the output, we use $\langle q \rangle \langle u \rangle$ or $\langle q \rangle \langle v \rangle$ to evaluate the moisture flux contribution to the region from each side of the domain, instead of $\langle qu \rangle$ or $\langle qv \rangle$[49].

**Fast and slow components**. According to refs. [56-58], we have also separated the climate responses (including AWJS position and precipitation) into fast and slow components. The total response ($\Delta S_{total}$) over the last 50 years is considered as a linear sum of the fast component and slow component in the Coupled simulations. The fast component of climate variable S due to rapid adjustments (e.g., atmospheric heating or cooling), $\Delta S_{fast}$, is obtained from the fSST numerical simulations over the last 10 years. Therefore, the slow component ($\Delta S_{slow}$) can be derived from

$$\Delta S_{slow} = \Delta S_{total} - \Delta S_{fast}, \qquad (2)$$

where the component $\Delta S_{slow}$ indicates the climate response due to the feedbacks induced by surface sea temperature.

## Data availability

The PDRMIP dataset of numerical simulations can be accessed online (http://www.cicero.uio.no/en/PDRMIP/PDRMIP-data-access). CRU TS4.04 is the gridded Climatic Research Unit Time-series data version 4.04 data, available through https://catalogue.ceda.ac.uk/uuid/89e1e34ec3554dc98594a5732622bce9. NCEP/NCAR Reanalysis 1 is obtained from NOAA/PSL through https://psl.noaa.gov/data/gridded/data.ncep.reanalysis.html.

## Code availability

All analyses were performed using the NCAR Command Language (NCL version 6.4.0). The plotting scripts for the figures are available from the corresponding author on request.

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

## Acknowledgements

This research has been supported by the National Natural Science Foundation of China (41991254) and the Strategic Priority Research Program of the Chinese Academy of Sciences (XDB40030100). X.N.X. is supported by the National Natural Science Foundation of China (42175059) and the CAS "Light of West China" program (XAB2019A02). Y.L. was supported by the US Department of Energy's Atmospheric System Research (ASR) program.

## Author contributions

X.N.X. and Y. L. produced the analysis and wrote the manuscript. G.M., D.S., G.F., T.T., A.V., Z.G.S., X.Z.L, X.X.X., H.L., and X.D.L. contributed to the commenting of the results and revising of the manuscript.

## Competing interests

The authors declare no competing interests.
