## [Peer Review File · Communications Earth & Environment]

5th Jul 22

Dear Dr Xie,

Your manuscript titled "Teleconnection-like effect on increase of summer precipitation over arid Central Asia by anthropogenic aerosols in heavily polluted regions" has now been seen by 3 reviewers, and I include their comments at the end of this message. They find your work of interest, but some important points are raised. We are interested in the possibility of publishing your study in Communications Earth & Environment, but would like to consider your responses to these concerns and assess a revised manuscript before we make a final decision on publication.

We therefore invite you to revise and resubmit your manuscript, along with a point-by-point response that takes into account the points raised. Specifically, your revision should critically address the following (a) provide compelling new insights into the drivers of precipitation trends over arid Central Asia, (b) demonstrate that your extremely idealized experiment (PDRMIP) is suitable and robust, and discuss/evaluate its performance in comparison to other approaches, for example CMIP6 (historical, hist-GHG and hist-aer), (c) provide an in-depth balanced discussion of the advance your study offers beyond the related literature, and (d) describe the underlying mechanism(s) linking the southward shifted jet stream and the increased ACA precipitation.

Please highlight all changes in the manuscript text file.

Please use the following link to submit your revised manuscript, point-by-point response to the referees' comments (which should be in a separate document to any cover letter) and the completed checklist:

[link redacted]

We hope to receive your revised paper within six weeks; please let us know if you aren't able to submit it within this time so that we can discuss how best to proceed. If we don't hear from you, and the revision process takes significantly longer, we may close your file. In this event, we will still be happy to reconsider your paper at a later date, as long as nothing similar has been accepted for publication at Communications Earth & Environment or published elsewhere in the meantime.

We understand that due to the current global situation, the time required for revision may be longer than usual. We would appreciate it if you could keep us informed about an estimated timescale for resubmission, to facilitate our planning. Of course, if you are unable to estimate, we are happy to accommodate necessary extensions nevertheless.

Please do not hesitate to contact me if you have any questions or would like to discuss these revisions further. We look forward to seeing the revised manuscript and thank you for the opportunity to review your work.

Best regards,

Akintomide Akinsanola, PhD
Editorial Board Member
Communications Earth & Environment

EDITORIAL POLICIES AND FORMATTING

Editorial Policy: [Policy requirements](https://www.nature.com/documents/nr-editorial-policy-checklist.zip)

Furthermore, please align your manuscript with our format requirements, which are summarized on the following checklist:

[Communications Earth & Environment formatting checklist](https://www.nature.com/documents/commsj-phys-style-formatting-checklist-article.pdf)

and also in our style and formatting guide [Communications Earth & Environment formatting guide](https://www.nature.com/documents/commsj-phys-style-formatting-guide-accept.pdf) .

*** DATA: Communications Earth & Environment endorses the principles of the Enabling FAIR data project (<http://www.copdess.org/enabling-fair-data-project/>). We ask authors to make the data that support their conclusions available in permanent, publically accessible data repositories. (Please contact the editor if you are unable to make your data available).

All Communications Earth & Environment manuscripts must include a section titled "Data Availability" at the end of the Methods section or main text (if no Methods). More information on this policy, is available at <http://www.nature.com/authors/policies/data/data-availability-statements-data-citations.pdf>.

- Unique identifiers (such as DOIs and hyperlinks for datasets in public repositories)
- Accession codes where appropriate
- If applicable, a statement regarding data available with restrictions

- If a dataset has a Digital Object Identifier (DOI) as its unique identifier, we strongly encourage including this in the Reference list and citing the dataset in the Data Availability Statement.

If a community resource is unavailable, data can be submitted to generalist repositories such as [figshare](https://figshare.com/) or [Dryad Digital Repository](http://datadryad.org/). Please provide a unique identifier for the data (for example a DOI or a permanent URL) in the data availability statement, if possible. If the repository does not provide identifiers, we encourage authors to supply the search terms that will return the data. For data that have been obtained from publically available sources, please provide a URL and the specific data product name in the data availability statement. Data with a DOI should be further cited in the methods reference section.

REVIEWER COMMENTS:

Reviewer #1 (Remarks to the Author):

Review of “Teleconnection-like effect on increase of summer precipitation over arid Central Asia by anthropogenic aerosols in heavily polluted regions”

This paper uses idealized single forcing experiments from equilibrium PDRMIP experiments to attribute 1961-2005 wetting of Arid Central Asia to aerosols (e.g., 10xBC/10xSO₄ over Asia). I do not see how extremely idealized model experiments can be used for such a purpose. I agree that they can be used to primarily understand mechanisms, and secondly to gain insight on climate responses. But I feel as if their utility for a formal attribution study is low. CMIP6 simulations are more appropriate for such purposes, and such experiments exist (e.g., historical, hist-GHG and hist-aer). Why are these more realistic simulations not the focus here? I guess this question is eventually answered...CMIP6 simulations do not yield similar results. The authors then argue this is due to a “low aerosol bias”. What about the incredibly large “high aerosol bias” in PDRMIP—the focus of this analysis and the basis for the authors conclusions?

The authors focus on the PDRMIP 10xSO₂/SO₄ Asia experiments, to suggest the aforementioned wetting is due to aerosols. What about emissions in other regions? What about (simultaneous/contemporaneous) emissions of other aerosol species? Are south Asian

aerosol emissions not important? Such aerosol/precursor gas emissions have also significantly increased (as have Asian aerosols...except for the last decade or so), and south Asia is also located near this study's focus area. Oddly, Fig. 5 (the cartoon) does show south Asian aerosols? How do they factor in?

I also note a similar paper published earlier this year:

Dong, B., Sutton, R.T., Shaffrey, L. et al. Recent decadal weakening of the summer Eurasian westerly jet attributable to anthropogenic aerosol emissions. *Nat Commun* 13, 1148 (2022). <https://doi.org/10.1038/s41467-022-28816-5>

The authors do cite this paper, but only casually. It would appear the two studies are highly related to one another? Is more discussion not warranted? Although this paper focuses more in precipitation, the proposed mechanisms are essentially the same. And Dong et al. (2022) does briefly address precipitation. It appears CMIP6 models (including aerosol-only simulations) do show some JJA wetting in the ACA.

I guess the authors do go on to briefly analyze CMIP5/6 models, but they find weaker results (and again, they suggest this is because CMIP5/6 models have a "low aerosol bias"). And it appears the authors are arguing the idealized, single forcing PDRMIP experiments are more accurate/better suited for an attribution study, such as this? Isn't PDRMIP's aerosol "biased high"? Like, very high (since the emissions/concentrations are multiplied by 10)? So couldn't one use the same argument the authors use for CMIP5/6 but reverse it, i.e., that PDRMIP overestimates the aerosol response because they use 10x emissions/concentrations? And again, what about south Asian aerosols?

L111. "This significant increase in ACA summer precipitation had not been identified in previous work (Menon et al., 2002; Ramanathan et al., 2005; Lau et al., 2008; Bollasina et al., 2011; Liu et al., 2011; Song et al., 2014; Li et al., 2016b; Xie et al., 2016; Shawki et al., 2018; Dong et al., 2019), due likely to substantial underestimations of anthropogenic aerosol burden and the corresponding AOD over Asia in those GCM studies (Shindell et al., 2013; Lamarque et al., 2013; Pan et al., 2015; Fan et al., 2018)" ^[11]_{SEP} Do these references also suggest the newest, CMIP6 models likewise underestimate Asian AOD? What about indirect aerosol effects? Are these not important? What about internal climate variability? Surely this has contributed, to some extent, to the observed changes?

Reviewer #2 (Remarks to the Author):

Summary of my review

This study used a series of numerical model experiments, and revealed the possible role of aerosol in the decadal wetting trend over Arid central Asia (ACA). Based on the model experiments designed by PDRMIP, the authors analyzed the effect of aerosol emission on the

precipitation, atmospheric circulation and the associated moisture budget. The authors proposed that the increased emission of anthropogenic (sulfate) aerosols over Asia is responsible for the observed decadal rising trend of precipitation over ACA region, by shifting the Asian westerly jet stream southward. Generally, this study is well designed, and most of the analysis and explanation are reasonable. The paper may be considered for publication after addressing the following three major issues and some minor issues.

Major issues

1) The authors well explained the mechanism connecting the southward shift of the jet stream, but didn't clearly explain why the southward shifted jet stream increases ACA precipitation. The authors stated that the southward shift of the upper tropospheric jet stream induces a low-level southwestward (southwesterly?) wind at lower levels and ascending motion over ACA. I don't understand why the southward shift of upper level jet stream induces a low-level southwesterly wind anomaly? The mechanisms need to be explained in detail.

2) The well-known "monsoon-desert" mechanism is not mentioned in this work. In fact, weakened ascent over Asian monsoon region may be directly responsible for the weakened descent over the arid region on its northwestern side, which could be an alternative explanation on the decadal wetting trend and associated low-level circulation.

Rodwell MJ, Hoskins BJ (1996) Monsoons and the dynamics of deserts. *Q J Roy Meteor Soc* 122 (534):1385-1404. doi:<https://doi.org/10.1002/qj.49712253408>

Kripalani RH, Oh JH, Kang JH, Sabade SS, Kulkarni A (2005) Extreme monsoons over East Asia: Possible role of Indian Ocean Zonal Mode. *Theor Appl Climatol* 82 (1-2):81-94. doi:DOI 10.1007/s00704-004-0114-z

3) In recent two years, aerosol emission is sharply suppressed due to COVID-19 pandemic, which has already reversed the decadal weakening trend of Asian monsoon rainfall (He et al. 2022; Kripalaniet al. 2022). I suggest the authors examine the observed averaged precipitation anomalies over ACA region in 2020 and 2021, and discuss whether the reduction in aerosol in recent two years has reversed the decadal wetting trend over ACA region. Could the anomalous precipitation over ACA region under aerosol reduction during COVID-19 pandemic also be explained by the mechanism you proposed?

He C, Zhou W, Li T, Zhou T, Wang Y (2022) East Asian summer monsoon enhanced by COVID-19. *Clim Dynam.* doi:10.1007/s00382-022-06247-8

Kripalani R, Ha K-J, Ho C-H, Oh J-H, Preethi B, Mujumdar M, Prabhu A (2022) Erratic Asian Summer Monsoon 2020: COVID-19 Lockdown Initiatives Possible Cause for These Episodes? *Clim Dynam.* doi:10.21203/rs.3.rs-839934/v1

Minor issues

1) Only the trend of upper-level circulation is shown in the main text. Since moisture concentrates in the boundary layer, I agree with the authors that the precipitation is the most directly connected to low-level circulation and moisture transport, rather than the upper troposphere. Therefore, I suggest the authors show the observed trend and simulated

response of low-level circulation in the main text, rather than in the supplementary information (Fig. S4).

2) Fig. S11 and L252-253: How many models have you used in Fig. S11? Based on the MMM of 30 models, He et al. (2019) showed that East Asian subtropical jet shifts southward in response to global warming, which is different from the shift of the zonal mean jet.

He C, Wang Z, Zhou T, Li T (2019) Enhanced Latent Heating over the Tibetan Plateau as a Key to the Enhanced East Asian Summer Monsoon Circulation under a Warming Climate. *J Climate* 32 (11):3373-3388. doi:10.1175/jcli-d-18-0427.1

3) L150, L172, L244: You may probably mean "southwesterly" wind, rather than "southwestward" (northeasterly) wind.

4) L31, L233: "wet-get-wetter" mechanism explains global-scale precipitation response to global warming, and it is not surprising that some regional features are inconsistent with "wet-get-wetter". So, the observed wetting trend over ACA region does not "challenge" the "wet-get-wetter" mechanism.

Reviewer #3 (Remarks to the Author):

Review of COMMSENV-22-0441-T: "Teleconnection-like effect on increases of summer precipitation over arid Central Asia by anthropogenic aerosols in heavily polluted areas"

This manuscript presents an analysis of the observed increase in precipitation that has occurred over arid Central Asia (ACA) during summer and winter, despite the well-known paradigm that the "dry get drier and the wet get wetter" as a result of global warming. The drivers of the ACA wetting have remained controversial and the Author's set out to get to the bottom of this debate. Through analysis of models from the Precipitation Driver and Response Model Intercomparison Project (PDRMIP), the Author's find that anthropogenic aerosols over remote Asian polluted regions, including the Indian subcontinent and eastern China, dominates the increase of ACA summer precipitation, specifically convective and extreme precipitation. The manuscript is structured to convince us of the mechanisms behind this. The Author's suggest that Asian anthropogenic aerosols cause an equatorward shift of the summertime Asian Westerly Jet (AWJS) through fast responses in response to cooling the local mid-latitude atmosphere which favors moisture supply from low latitude and moisture flux convergence of the ACA region. This is found through analysis of sensitivity experiments in PDRMIP where there is either a tenfold increase in present-day sulfate concentrations across the Asia region, a tenfold increase in black carbon aerosols over the Asia region, or doubled CO₂. In summary, the Author's suggest that this teleconnection between remote Asian polluted region aerosols and the ACA wetting highlights the long-range impacts of anthropogenic aerosols on atmospheric circulations and hydroclimate.

This paper is interesting to read, clear, and informative and I recommend this paper for publication after some minor revisions outlined below. I find the work to be novel in

investigating a precipitation increase in a region of the world where we would expect a decrease in precipitation based on the paradigm of “the dry get drier and the wet get wetter”. I think this manuscript will be interesting for the community and wider field in exploring regional precipitation trends.

Specific comments

Introduction

L69: The Author’s note that the drivers of ACA regional wetting remain highly controversial during the summer. If there is space, it would be interesting to expand a bit more about why summer and perhaps the relevant seasonality of the AWJS.

L72: I think that you could just write ‘increasing GHG by doubling CO₂’.

L72-73: The Author’s only note the increase in winter precipitation over ACA in Figure 1d but what about the increase in spring precipitation? Is it because the error bars are much larger and more negative for spring compared to summer?

L74: Here or somewhere within this description of the spatial trend of the JJA U200 trend it would be good to point the reader to Figure 1c again, I think.

L97: I think it might be worth adding a box to some of the supplemental figures that show the region bounds for the SULx10Asia and BCx10Asia sensitivity experiments.

Results

Figs. S1, S2, S4: Perhaps it would be helpful to add a box over the ACA region in at least one or all of these figures and any of the other supplemental regions with maps.

L108: I wonder if there is space to explain a bit more about how Asian sulfate aerosols reduce summertime precipitation through weakening the South and East Asian summer monsoons. This would provide more context for the readers.

L110: I don’t think you need to have arid in front of ACA since arid is in the acronym. This pops up in a few other places as well.

L122: In the parentheses it might be good to add some text for clarity so it reads ‘(larger than 65% of total precipitation)’ or something like this.

L122-123: I think that ‘with’ can be deleted and the sentence can read ‘Accompanying the increase in convective precipitation there is a sizeable enhancement of 7.4% also shown in the extreme precipitation’.

L137: At the end of this sentence the Authors could point to Figures 3a and 3b.

Figure 3: In panel b the text in the figure above the bars is quite distracting. Is it really necessary there? Or in general do you need these values within the main text?

L150: I believe that 'southwestward wind' should actually be 'southwesterly winds' since the winds are coming from the southwest if I am following correctly.

L165: Same comment as L110

L172: Same comment as L150

L177: It might be good to add to the caption of Figure S6 that these results are from the SULx10Asia experiments for consistency in wording. I think that is what is meant by 'Asian sulfate-induced changes...' and so the Authors could just add 'Asian sulfate-induced changes (SULx10Asia)...' if that is the case.

L196: Where the Authors point to Fig. S8 could point directly to Fig. S8b to be consistent with pointing to Fig. S3e in the same sentence.

L198-201: This sentence could be re-worded for clarity. Specifically, I think that discover could be replaced with investigate.

L224: Same comment as L72-73, what about the spring increase in Figure 1d?

L226: Where the Authors point to Fig. S10, it might be good to point directly to the panels referring to which in this case I believe should be Figs. S10b and S10d.

Concluding remarks

L233: I think 'region' should be included after '(ACA)'

L240: I think this sentence can be re-worded to read '...(AWJS) to shift equatorward'

L244: I think 'southwestward' should be replaced with 'southwesterly'. Also, I think there should be a 'the' between 'by moisture' so it reads as 'by the moisture budget analysis'

L254: Is it obvious to a broad audience? I encourage the Author's to maybe re-word this.

Methods

L280: It is not so clear to me what is meant by BASE. If this is an acronym, I think it should be spelt out or if it is just how the Authors are referring to the historical simulations with no modifications/sensitivity testing, I think that should be explained more clearly.

L284: Why were CanESM2 and HadGEM2 left out?

L285-287: Does '15 year running' and '100 year running' mean that these are 15- and 100-year runs? This is a little confusing to me and I wonder if it can be re-worded for clarity.

Perhaps something like:

In the PDRMIP experiments, a pair of numerical simulations was conducted for the fixed sea surface temperature (fSST; run for 15 years) and the coupled climate simulations (Coupled; run for 100 years).

L295: I think the sentence should start 'Each pair of models...'

L317: What is the temporal resolution of the data used for the moisture budget analysis?
More broadly it might be good to state the temporal resolution of all data considered within the Methods section.

Response to Reviewer #1:

(1), This paper uses idealized single forcing experiments from equilibrium PDRMIP experiments to attribute 1961-2005 wetting of Arid Central Asia to aerosols (e.g., 10xBC/10xSO₄ over Asia). I do not see how extremely idealized model experiments can be used for such a purpose. I agree that they can be used to primarily understand mechanisms, and secondly to gain insight on climate responses. But I feel as if their utility for a formal attribution study is low. CMIP6 simulations are more appropriate for such purposes, and such experiments exist (e.g., historical, hist-GHG and hist-aer). Why are these more realistic simulations not the focus here? I guess this question is eventually answered...CMIP6 simulations do not yield similar results. The authors then argue this is due to a “low aerosol bias” . What about the incredibly large “high aerosol bias” in PDRMIP-the focus of this analysis and the basis for the authors conclusions?

Response: Thank you very much for the constructive comments, which are useful to revise and improve our manuscript significantly.

We absolutely agree with the Reviewer’s point about the formal attribution study used in CMIP5 and CMIP6 multi-models. Note that substantial underestimations of anthropogenic aerosols have been reported over Asia in CMIP5 models (Shindell et al., 2013; Lamarque et al., 2013; Pan et al., 2015; Fan et al., 2018), which is also shown in Fig. R1d from the reference (Li et al., 2021). The latest CMIP6 models show incremental improvements in the simulation of anthropogenic aerosols compared to CMIP5 models (Cherian and Quaas, 2020; Li et al., 2021; Ramachandran et al., 2022), especially over Asia including eastern China (Fig. R1d, Li et al., 2021) and the Indian subcontinent (Ramachandran et al., 2022), although they also underestimate AOD over Asia (Fig1d; Cherian and Quaas, 2020; Li et al., 2021; Ramachandran et al., 2022). However, compared with observations (Fig. R2a), the CMIP6 models in the hist-aer experiments only show the weakening of the summer Asian Westerly Jet Stream (AWJS) and do not reproduce its southward shift (in Fig. R2c, from Dong et al., 2020), also shown in our results with different eight CMIP6 models in Fig. R3. Furthermore, these CMIP6 models only show a JJA precipitation increase over the mountains in the ACA region (Fig. R2d, from Dong et al., 2020), whereas the observations show a JJA precipitation increase over the whole region, especially over Northwest China (Fig. R2b).

Here, we used these idealized single forcing experiments (e.g., CO₂x2, SULx10Asia, BCx10Asia) in PDRMIP. The PDRMIP results indeed show new insights in the southward shift of AWJS and the precipitation increase over the whole ACA region in Fig. R4, which is very similar with the observations (Figs. R2a and R2b). Although these results are based on sensitivity experiments with high aerosol bias

and do not show a formal attribution study, they can help us understand mechanisms and gain insight on climate responses over the ACA region. Together with our study, these results of sensitivity experiments suggest the need to quantify the relative contributions of the different factors (e.g., anthropogenic aerosols, greenhouse gases, and internal forcings) to the ACA wetting. Therefore, we have revised the corresponding descriptions in the revised manuscript.

Fig. R1 (from Fig. 1 in Li et al. 2021) Spatial distributions of the multi-year annual mean AOD during the period of 2000 to 2005 by (a) MODIS onboard Terra, (b) MODIS onboard Aqua, (c) CMIP6 multi-model ensemble (MME), and (d) CMIP5 MME. Filled circles in (a) represent the 2000–2005 surface observational AOD decade at selected AERONET sites. The red rectangle (103–123E, 22–41N) denotes the area of East-Central China defined in this study. The two blue rectangles (north: 110–123E, 22–30N; south: 110–123E, 30–41N) denote the two sub-regions North China (NC) and South China (SC).

(a) U200 trend (observations)

(b) Precipitation trend (observations)

(c) U200 trend (CMIP6)

(d) Precipitation (CMIP6)

Fig. R2 Spatial pattern in trends of (a) JJA westerlies at 200 hPa (U200, m s⁻¹ per 100 years) and (b) JJA precipitation (mm day⁻¹ per 100 years) in observations from 1961 to 2005. But (c) JJA U200 and (d) JJA precipitation in CMIP6 simulations (from Figs. 2e and 4i in Dong et al., 2022).

Fig. R3 Comparisons of trends of JJA Asian jet position (degree per 100 years) and ACA precipitation ($10^{-1}\text{mm day}^{-1}$ per 100 years) during 1961-2005 between observations (NCEP/NCAR Reanalysis 1 and CRU TS4.04) and CMIP5/CMIP6 multi-model mean (MMM). Error bars of MMM represent the standard deviation among the CMIP5/CMIP6 models, which is based on 8 CMIP5 models and 8 CMIP6 models detailedly described by Xin et al. (2020).

Fig. R4 Asian sulfate-induced changes in JJA Asian westerlies and JJA precipitation over arid Central Asia (ACA). Slanted lines indicate where the MMM is more than 1 standard deviation away from 0 and red box (b) indicates the ACA region.

(2), The authors focus on the PDRMIP 10xSO₂/SO₄ Asia experiments, to suggest the aforementioned wetting is due to aerosols. What about emissions in other regions? What about (simultaneous/contemporaneous) emissions of other aerosol species? Are south Asian aerosol emissions not important? Such aerosol/precursor gas emissions have also significantly increased (as have Asian aerosols...except for the last decade or so), and south Asia is also located near this study's focus area. Oddly, Fig. 5 (the cartoon) does show south Asian aerosols? How do they factor in?

Response: Thank you very much for the constructive comments. As mentioned in Samset et al., 2016; Myhre et al., 2017, and Liu et al. 2018, the sensitivity experiments of a tenfold increase in present-day sulfate concentrations or emissions only across the Asia region (60°E–140°E, 10°N–50°N) are conducted in PDRMIP in SULx10Asia (Table R1), whereas other aerosol species are fixed as present-day concentrations or emissions. The sensitivity experiments of BCx10Asia are similar with SULx10Asia but for black carbon aerosols.

Table R1 Model simulations analyzed in the current study. The scaling refers to concentrations, but in two models (MIROC-SPRINTARS and CESM1-CAM5) the corresponding anthropogenic emissions were increased by 10 times.

Experiments	CO ₂ x2	SULx10Asia	BCx10Asia
Specifications	doubled CO ₂	SO ₄ over Asia increased by 10 times	BC over Asia increased by 10 times

The reviewer's point is correct. South Asia is located near this study's focus area (Central Asia), where such aerosol/precursor gas emissions have also significantly increased in Fig. S1b. It is noted that the sensitivity experiments of SULx10Asia and BCx10Asia are across the whole Asia region (60°E–140°E, 10°N–50°N), which absolutely includes the south Asia. In Figs. S7a and S7b, it shows a significant increase in AOD and ERF over South Asia and eastern China in the SULx10Asia experiments.

(a) MERRA-2 AOD

(b) MERRA-2 AOD trend

Fig. S1 (a), Spatial pattern of annual mean aerosol optical depth (AOD) from 1980 to 2005 and (b), for annual AOD trend (per 100 years) during the same period based on MERRA-2 aerosol data, available through <https://disc.gsfc.nasa.gov/datasets?project=MERRA-2>. Red box in a and b indicates the ACA region and slanted lines (as shown in b) represent significance at the 95% confidence level by a standard t-test.

Fig. S7 Asian sulfate-induced changes (SULx10Asia) of the multi-model mean (MMM) in (a) JJA aerosol optical depth (AOD), (b) effective radiative forcing (ERF, W m^{-2}), (c) zonally mean temperature (T, $^{\circ}\text{C}$), and (d) zonal mean temperature gradient (MGT, $10^{-3} \text{ }^{\circ}\text{C km}^{-1}$). Thin black lines show the climatological temperature (c) and temperature gradient (d). Slanted lines represent where MMM is more than 1 standard deviation of the PDRMIP models away from 0.

(3), I also note a similar paper published earlier this year:

Dong, B., Sutton, R.T., Shaffrey, L. et al. Recent decadal weakening of the summer Eurasian westerly jet attributable to anthropogenic aerosol emissions. *Nat Commun* 13, 1148 (2022). <https://doi.org/10.1038/s41467-022-28816-5>

The authors do cite this paper, but only casually. It would appear the two studies are highly related to one another? Is more discussion not warranted? Although this paper focuses more in precipitation, the proposed mechanisms are essentially the same. And Dong et al. (2022) does briefly address precipitation. It appears CMIP6 models (including aerosol-only simulations) do show some JJA wetting in the ACA.

Response: Yes, the paper (Dong et al., 2022) is very important and interesting, which also supports our main results. We have added more discussions about a comparison with the corresponding CMIP6 results. Substantial underestimations of anthropogenic aerosols have been reported over Asia in CMIP5 models (Shindell et al., 2013; Lamarque et al., 2013; Pan et al., 2015; Fan et al., 2018), which is also shown in Fig. R1 from the reference (Li et al., 2021). Note that the latest CMIP6 models show incremental improvements in the simulation of anthropogenic aerosols compared to CMIP5 models (Cherian and Quaas, 2020; Li et al., 2021; Ramachandran et al., 2022), especially over Asia including eastern China (Fig. R1, Li et al., 2021) and the Indian subcontinent (Ramachandran et al., 2022), although they also underestimate AOD over Asia (Cherian and Quaas, 2020; Li et al., 2021; Ramachandran et al., 2022). Note that, using the latest CMIP6 models, Dong et al. 2022 shows that anthropogenic aerosols were a potential cause of the changing subtropical westerly jet over recent decades and also implies the regional precipitation increase over ACA in the hist-aer experiments. These CMIP6 results also support our proposed mechanism of aerosol-induced increase in summer precipitation over the ACA region through affecting AWJS.

Fig. R1 (from Fig. 1 in Li et al. 2021) Spatial distributions of the multi-year annual mean AOD during the period of 2000 to 2005 by (a) MODIS onboard Terra, (b) MODIS onboard Aqua, (c) CMIP6 multi-model ensemble (MME), and (d) CMIP5 MME. Filled circles in (a) represent the 2000–2005 surface observational AOD decade at selected AERONET sites. The red rectangle (103–123E, 22–41N) denotes the area of East-Central China defined in this study. The two blue rectangles (north: 110–123E, 22–30N; south: 110–123E, 30–41N) denote the two sub-regions North China (NC) and South China (SC).

(4), I guess the authors do go on to briefly analyze CMIP5/6 models, but they find weaker results (and again, they suggest this is because CMIP5/6 models have a “low aerosol bias”). And it appears the authors are arguing the idealized, single forcing PDRMIP experiments are more accurate/better suited for an attribution study, such as this? Isn't PDRMIP's aerosol “biased high” ? Like, very high (since the emissions/concentrations are multiplied by 10)? So couldn't one use the same argument the authors use for CMIP5/6 but reverse it, i.e., that PDRMIP overestimates the aerosol response because they use 10x emissions/concentrations? And again, what about south Asian aerosols?

Response: As mentioned in Samset et al., 2016; Myhre et al., 2017, and Liu et al. 2018, the sensitivity experiments of a tenfold increase in present-day sulfate concentrations or emissions only across the Asia region (60°E–140°E, 10°N–50°N) are conducted in PDRMIP in SULx10Asia (Table R1), whereas other aerosol species are fixed as present-day concentrations or emissions. We absolutely agree with the Reviewer's point about the formal attribution study used in CMIP5 and CMIP6 multi-models. Here, we used idealized single forcing experiments (e.g., CO2x2, SULx10Asia, BCx10Asia) in PDRMIP to understand mechanisms and gain insight on climate responses over the arid Central Asia (ACA). Therefore, we only provide an underlying mechanism to understand the ACA summer wetting in our manuscript, but do not show a formal attribution study. Together with our study, these results of sensitivity experiments suggest the need to quantify the relative contributions of the different factors (e.g., anthropogenic aerosols, greenhouse gases, and internal forcings) to the ACA wetting.

Table R1 Model simulations analyzed in the current study. The scaling refers to concentrations, but in two models (MIROC-SPRINTARS and CESM1-CAM5) the corresponding anthropogenic emissions were increased by 10 times.

Experiments	CO2x2	SULx10Asia	BCx10Asia
Specifications	doubled CO2	SO4 over Asia increased by 10 times	BC over Asia increased by 10 times

Additionally, South Asia is located near this study's focus area (Central Asia), where such aerosol/precursor gas emissions have also significantly increased in Fig. S1. It is noted that the sensitivity experiments of SULx10Asia and BCx10Asia are across the whole the Asia region (60°E–140°E, 10°N–50°N), which absolutely includes the south Asia. In Figs. S7a and S7b, it shows a significant increase in AOD and ERF over South Asia and eastern China in the SULx10Asia experiments.

Fig. S7 Asian sulfate-induced changes (SULx10Asia) of the multi-model mean (MMM) in (a) JJA aerosol optical depth (AOD), (b) effective radiative forcing (ERF, W m^{-2}), (c) zonally mean temperature (T , $^{\circ}\text{C}$), and (d) zonal mean temperature gradient (MGT, $10^{-3} \text{ }^{\circ}\text{C km}^{-1}$). Thin black lines show the climatological temperature (c) and temperature gradient (d). Slanted lines represent where MMM is more than 1 standard deviation of the PDRMIP models away from 0.

(5), L111. “This significant increase in ACA summer precipitation had not been identified in previous work (Menon et al., 2002; Ramanathan et al., 2005; Lau et al., 2008; Bollasina et al., 2011; Liu et al., 2011; Song et al., 2014; Li et al., 2016b; Xie et al., 2016; Shawki et al., 2018; Dong et al., 2019), due

likely to substantial underestimations of anthropogenic aerosol burden and the corresponding AOD over Asia in those GCM studies (Shindell et al., 2013; Lamarque et al., 2013; Pan et al., 2015; Fan et al., 2018). Do these references also suggest the newest, CMIP6 models likewise underestimate Asian AOD? What about indirect aerosol effects? Are these not important? What about internal climate variability? Surely this has contributed, to some extent, to the observed changes?

Response: Substantial underestimations of anthropogenic aerosols have been reported over Asia in CMIP5 models (Shindell et al., 2013; Lamarque et al., 2013; Pan et al., 2015; Fan et al., 2018), which is also shown in Fig. R1 from the reference (Li et al., 2021). The latest CMIP6 models show incremental improvements in the simulation of anthropogenic aerosols compared to CMIP5 models (Cherian and Quaas, 2020; Li et al., 2021; Ramachandran et al., 2022), especially over Asia including eastern China (Fig. R1, Li et al., 2021) and the Indian subcontinent (Ramachandran et al., 2022), although they also underestimate AOD over Asia (Cherian and Quaas, 2020; Li et al., 2021; Ramachandran et al., 2022). Note that, using the latest CMIP6 models, Dong et al. 2022 shows that anthropogenic aerosols were a potential cause of the changing subtropical westerly jet over recent decades and also implies the regional precipitation increase over ACA in the hist-aer experiments. These CMIP6 results support our proposed mechanism of aerosol-induced increase in summer precipitation over the ACA region through affecting AWJS.

Table R2 shows the model descriptions of the seven GCMs used here. All the GCMs include the aerosol indirect effect except GISS-E2-R and NCAR-CESM1-CAM4. Asian anthropogenic aerosols cause an equatorward shift of the summer Asian Westerly Jet Stream (AWJS) through fast responses due to cooling the local atmosphere in mid-latitudes. It is noted that the cooling of the local atmosphere in mid-latitudes is mainly attributed to both the aerosol direct and indirect effects. Additionally, here we only provide an underlying mechanism to understand the ACA summer wetting in our manuscript, but do not show a formal attribution study. Together with our study, these results of sensitivity experiments suggest the need to quantify the relative contributions of the different factors (e.g., anthropogenic aerosols, greenhouse gases, and internal forcings) to the ACA wetting.

Fig. R1 (from Fig. 1 in Li et al. 2021) Spatial distributions of the multi-year annual mean AOD during the period of 2000 to 2005 by (a) MODIS onboard Terra, (b) MODIS onboard Aqua, (c) CMIP6 multi-model ensemble (MME), and (d) CMIP5 MME. Filled circles in (a) represent the 2000–2005 surface observational AOD decade at selected AERONET sites. The red rectangle (103–123E, 22–41N) denotes the area of East-Central China defined in this study. The two blue rectangles (north: 110–123E, 22–30N; south: 110–123E, 30–41N) denote the two sub-regions North China (NC) and South China (SC).

Table R2: Models used for the present study as summarized in Myhre et al., (2017).

Model	Version	Horizontal resolutions	Vertical resolutions	Aerosol emissions	Indirect effects
GISS-E2-R	E2-R	2x2.5	40 levels	Fixed concentration	Sulfate and BC: no indirect effects
HadGEM3	GA 4.0	1.875x1.25	85 levels	Fixed concentration	Sulfate: all indirect effects; BC: no indirect effects
IPSL-CM5A	5A	3.75x1.875	19 levels	Fixed concentration	Sulfate and BC: first indirect effect
MIROC-SPRINTARS	5.9.0	1.4x1.4	40 levels	Emissions	Sulfate and BC: all indirect effects
NCAR-CESM1-CAM4	1.0.3	2.5x1.9	26 levels	Fixed concentration	Sulfate and BC: no indirect effects
NCAR-CESM1-CAM5	1.1.2	2.5x1.9	30 levels	Emissions	Sulfate and BC: all indirect effects
NorESM1	1-M	2.5x1.9	26 levels	Fixed concentrations	Sulfate and BC: all indirect effects

Response to Reviewer #2:

Summary of my review

This study used a series of numerical model experiments, and revealed the possible role of aerosol in the decadal wetting trend over Arid central Asia (ACA). Based on the model experiments designed by PDRMIP, the authors analyzed the effect of aerosol emission on the precipitation, atmospheric circulation and the associated moisture budget. The authors proposed that the increased emission of anthropogenic (sulfate) aerosols over Asia is responsible for the observed decadal rising trend of precipitation over ACA region, by shifting the Asian westerly jet stream southward. Generally, this study is well designed, and most of the analysis and explanation are reasonable. The paper may be considered for publication after addressing the following three major issues and some minor issues.

Response: Thank you very much for the positive and constructive comments to improve the manuscript significantly. We have addressed all the specific comments in the revised manuscript, with the point-by-point responses detailed below.

Major issues

1) The authors well explained the mechanism connecting the southward shift of the jet stream, but didn't clearly explain why the southward shifted jet stream increases ACA precipitation. The authors stated that the southward shift of the upper tropospheric jet stream induces a low-level southwestward (southwesterly?) wind at lower levels and ascending motion over ACA. I don't understand why the southward shift of upper level jet stream induces a low-level southwesterly wind anomaly? The mechanisms need to be explained in detail.

Response: We have added a new figure (Fig. S8) about the underlying mechanism linking the southward shifted jet stream, the low-level circulation change. Mid to upper-level atmospheric cooling leads to local a significant decrease in 200 hPa geopotential height over the mid-latitude regions, as shown in Fig. S8a. The significant decreases in atmospheric temperature and geopotential height induce a strong cyclonic anomaly in upper levels (Fig. S8b). The cyclonic anomaly in upper levels results in a southwesterly wind from upper to low levels such as the upper level (Fig. S8b), middle level (Fig. S8c), and low level (Fig. S8d).

Additionally, the relationships between the AWJS position and low-level circulation were also evident in interannual scales and inter-decadal scales. Based on recent observational studies, at the interannual timescale, there exists anomalous southerly winds and much more summer precipitation due to enhanced water vapor transport at the years with further south position of AWJS, compared with north position

(Zhao et al., 2014b; Du et al., 2016; Wei et al., 2017). Additionally, the inter-decadal variability of the AWJS shows a significant southward shift from the Reanalysis (shown in Fig. 1c) and also induces low-level southerly winds (Fig. S5), which is closely related to the summer wetting trend during recent decades over ACA (Zhao et al., 2014a; Peng and Zhou, 2017; Peng et al., 2018). Therefore, the AWJS-induced changes in low-level circulation result in more summer convective precipitation and also more extreme precipitation over the ACA region, as shown in Fig. 2.

Fig. S8 Asian sulfate-induced changes (SULx10Asia) of the multi-model mean (MMM) in (a) JJA 200 hPa geopotential height (Zg200, gpm), JJA 200 hPa wind field (UV200, m s⁻¹), JJA 500 hPa wind field (UV500, m s⁻¹), and JJA 700 hPa wind field (UV700, m s⁻¹).

2) The well-known "monsoon-desert" mechanism is not mentioned in this work. In fact, weakened ascent over Asian monsoon region may be directly responsible for the weakened descent over the arid region on its northwestern side, which could be an alternative explanation on the decadal wetting trend and associated low-level circulation.

Rodwell MJ, Hoskins BJ (1996) Monsoons and the dynamics of deserts. *Q J Roy Meteor Soc* 122 (534):1385-1404. doi:<https://doi.org/10.1002/qj.49712253408>

Kripalani RH, Oh JH, Kang JH, Sabade SS, Kulkarni A (2005) Extreme monsoons over East Asia: Possible role of Indian Ocean Zonal Mode. *Theor Appl Climatol* 82 (1-2):81-94. doi:DOI 10.1007/s00704-004-0114-z

Response: Thank you very much for the constructive comments. Here, we used the equatorward shift of the summer Asian Westerly Jet Stream induced by Asian anthropogenic aerosols to explain the ACA wetting. This mechanism has been confirmed in observations the interannual timescale (Zhao et al., 2014b; Du et al., 2016; Wei et al., 2017), as well as during recent decades (Zhao et al., 2014a; Peng and Zhou, 2017; Peng et al., 2018). As mentioned by the Reviewer, the well-known "monsoon-desert" mechanism was proposed, claiming that the weakened ascent over the Asian monsoon region was directly responsible for the weakened descent over the ACA region (Rodwell et al., 1996; Kripalani et al., 2005). This mechanism can also explain the ACA wetting induced by Asian anthropogenic aerosols due to the weakening Asian monsoon. We have added the corresponding decrepitations in our manuscript.

3) In recent two years, aerosol emission is sharply suppressed due to COVID-19 pandemic, which has already reversed the decadal weakening trend of Asian monsoon rainfall (He et al. 2022; Kripalaniet al. 2022). I suggest the authors examine the observed averaged precipitation anomalies over ACA region in 2020 and 2021, and discuss whether the reduction in aerosol in recent two years has reversed the decadal wetting trend over ACA region. Could the anomalous precipitation over ACA region under aerosol reduction during COVID-19 pandemic also be explained by the mechanism you proposed?

He C, Zhou W, Li T, Zhou T, Wang Y (2022) East Asian summer monsoon enhanced by COVID-19. *Clim Dynam.* doi:10.1007/s00382-022-06247-8

Kripalani R, Ha K-J, Ho C-H, Oh J-H, Preethi B, Mujumdar M, Prabhu A (2022) Erratic Asian Summer Monsoon 2020: COVID-19 Lockdown Initiatives Possible Cause for These Episodes? *Clim Dynam.* doi:10.21203/rs.3.rs-839934/v1

Response: It is a very good suggestion to check the proposed mechanism using 2020 and 2021 due to COVID-19 pandemic. Fig. S13 shows a significant increase in the summer monsoon rainfall over South Asia and eastern China under aerosol reduction during COVID-19 pandemic (the years 2020 and 2021), as noted by He et al. 2022; Kripalaniet al. 2022, and a significant decrease in JJA precipitation over the ACA region. Therefore, the reduction of anthropogenic aerosols due to COVID-19 pandemic indeed

induces the decrease of JJA precipitation over ACA, which support the results and the proposed mechanisms. We also added these results in the discussions to make the result more credible.

Fig. S13 Observed JJA precipitation anomalies (%) for the years 2020 (a) and 2021 (b). The anomaly of precipitation is calculated by removing the climatology for the period 1979–2021 based on CRU TS Version 4.06.

Minor issues

1) Only the trend of upper-level circulation is shown in the main text. Since moisture concentrates in the boundary layer, I agree with the authors that the precipitation is the most directly connected to low-level circulation and moisture transport, rather than the upper troposphere. Therefore, I suggest the authors show the observed trend and simulated response of low-level circulation in the main text, rather than in the supplementary information (Fig. S4).

Response: Thanks very much for the Review’s suggestions. In the supplementary information, we also added the observed trend of the low-level circulation (Fig. S5). Due to the limit of the figure number in the main text, we just left these figures about the low-level circulation in observations (Fig. S5) and models (Fig. S4) in the supplementary information.

Fig. S5 Spatial pattern in trends of (a) JJA westerly winds and (b) JJA southerly winds at 700 hPa (m s^{-1} per 100 years) from 1961 to 2005 (NCEP/NCAR Reanalysis 1). Slanted lines represent significance at the 95% confidence level.

2) Fig. S11 and L252-253: How many models have you used in Fig. S11? Based on the MMM of 30 models, He et al. (2019) showed that East Asian subtropical jet shifts southward in response to global warming, which is different from the shift of the zonal mean jet.

He C, Wang Z, Zhou T, Li T (2019) Enhanced Latent Heating over the Tibetan Plateau as a Key to the Enhanced East Asian Summer Monsoon Circulation under a Warming Climate. *J Climate* 32 (11):3373-3388. doi:10.1175/jcli-d-18-0427.1

Response: We used eight CMIP5 models and eight CMIP6 models (Xin et al., 2020) with different horizontal and vertical resolutions in Section Methods. It is noted that we use the period of 1965-2005 in the CMIP5 historical simulations. In the previous studies (e.g., Song et al., 2014), they also claimed that Asian subtropical jet does not significantly changes during recent decades using 17 CMIP5 models, as similar with our results. However, He et al. (2019) pointed out that the westerly jet on the northern side of the TP shifts southward under global warming, using the 2050–99 periods in the RCP8.5 and RCP4.5 experiments. Note that these two results are different based on different time periods, which also indicates that responses of the westerly jet to global warming are very complex.

3) L150, L172, L244: You may probably mean "southwesterly" wind, rather than "southwestward" (northeasterly) wind.

Response: Taken

4) L31, L233: "wet-get-wetter" mechanism explains global-scale precipitation response to global warming, and it is not surprising that some regional features are in-consistent with "wet-get-wetter". So, the observed wetting trend over ACA region does not "challenge" the "wet-get-wetter" mechanism.

Response: Thanks very much. The paradigm of “dry gets drier, wet gets wetter” under global warming is a whole concept as proposed, the trend of dry to wetter over ACA challenge “dry gets drier” of this paradigm.

Response to Reviewer #3:

This manuscript presents an analysis of the observed increase in precipitation that has occurred over arid Central Asia (ACA) during summer and winter, despite the well-known paradigm that the “dry get drier and the wet get wetter” as a result of global warming. The drivers of the ACA wetting have remained controversial and the Author’ s set out to get to the bottom of this debate. Through analysis of models from the Precipitation Driver and Response Model Intercomparison Project (PDRMIP), the Author’ s find that anthropogenic aerosols over remote Asian polluted regions, including the Indian subcontinent and eastern China, dominates the increase of ACA summer precipitation, specifically convective and extreme precipitation. The manuscript is structured to convince us of the mechanisms behind this. The Author’ s suggest that Asian anthropogenic aerosols cause an equatorward shift of the summertime Asian Westerly Jet (AWJS) through fast responses in response to cooling the local mid-latitude atmosphere which favors moisture supply from low latitude and moisture flux convergence of the ACA region. This is found through analysis of sensitivity experiments in PDRMIP where there is either a tenfold increase in present-day sulfate concentrations across the Asia region, a tenfold increase in black carbon aerosols over the Asia region, or doubled CO₂. In summary, the Author’ s suggest that this teleconnection between remote Asian polluted region aerosols and the ACA wetting highlights the long-range impacts of anthropogenic aerosols on atmospheric circulations and hydroclimate.

This paper is interesting to read, clear, and informative and I recommend this paper for publication after some minor revisions outlined below. I find the work to be novel in investigating a precipitation increase in a region of the world where we would expect a decrease in precipitation based on the paradigm of “the dry get drier and the wet get wetter” . I think this manuscript will be interesting for the community and wider field in exploring regional precipitation trends.

Response: Thank you very much for the positive and constructive comments to improve the manuscript significantly. We have addressed all the specific comments in the revised manuscript, with the point-by-point responses detailed below.

Specific comments

Introduction

L69: The Author’ s note that the drivers of ACA regional wetting remain highly controversial during the summer. If there is space, it would be interesting to expand a bit more about why summer and perhaps the relevant seasonality of the AWJS.

Response: Thank you very much for the constructive comments. We have added the different mechanisms about the summer precipitation increase to expand this point. As shown in Fig. 1d, increasing GHGs by doubling CO₂ (labeled by CO₂x2) only result in the winter and spring precipitation increase over ACA, but do not lead to a significant AWJS equatorward shift or to a regional precipitation increase in summer (Xie et al., 2020), based on the multi-model results from the Precipitation Driver and Response Model Intercomparison Project (PDRMIP, see Methods).

L72: I think that you could just write ‘increasing GHG by doubling CO₂’ .

Response: Taken.

L72-73: The Author’ s only note the increase in winter precipitation over ACA in Figure 1d but what about the increase in spring precipitation? Is it because the error bars are much larger and more negative for spring compared to summer?

Response: Fig. 1d shows the seasonal changes induced by greenhouse gases (CO₂x2) in precipitation based on the sensitivity experiments for the multi-model mean (MMM) in PDRMIP models. We have added the spring precipitation in the descriptions. Fig. 1d shows a significant increase in precipitation during the winter, followed by the spring.

L74: Here or somewhere within this description of the spatial trend of the JJA U200 trend it would be good to point the reader to Figure 1c again, I think.

Response: Yes, we have added the descriptions about the spatial trend of JJA U200 in Fig. 1c.

L97: I think it might be worth adding a box to some of the supplemental figures that show the region bounds for the SULx10Asia and BCx10Asia sensitivity experiments.

Response: Thanks. Figs. S7a and S7b can obviously show the box mainly including South Asia and eastern China for the SULx10Asia and BCx10Asia sensitivity experiments.

Fig. S7 Asian sulfate-induced changes (SULx10Asia) of the multi-model mean (MMM) in (a) JJA aerosol optical depth (AOD), (b) effective radiative forcing (ERF, W m^{-2}), (c) zonally mean temperature (T, $^{\circ}\text{C}$), and (d) zonal mean temperature gradient (MGT, $10^{-3} \text{ }^{\circ}\text{C km}^{-1}$). Thin black lines show the climatological temperature (c) and temperature gradient (d). Slanted lines represent where MMM is more than 1 standard deviation of the PDRMIP models away from 0.

Results

Figs. S1, S2, S4: Perhaps it would be helpful to add a box over the ACA region in at least one or all of these figures and any of the other supplemental regions with maps.

Response: Taken.

L108: I wonder if there is space to explain a bit more about how Asian sulfate aerosols reduce summertime precipitation through weakening the South and East Asian summer monsoons. This would provide more context for the readers.

Response: Thanks for your useful suggestions. We have added the proposed mechanisms about the weakened the South and East Asian summer monsoon e.g., the reduction in the land-sea thermal gradients induced by aerosol-radiation and aerosol-cloud-radiation, as well as the second indirect aerosol effect (Ramanathan et al., 2005; Bollasina et al., 2011; Song et al., 2014; Li et al., 2016b; Xie et al., 2016; Shawki et al., 2018; Dong et al., 2019). Therefore, we have revised the description as “Consistent with previous studies with global climate models (GCMs) (Ramanathan et al., 2005; Bollasina et al., 2011; Song et al., 2014; Li et al., 2016b; Xie et al., 2016; Shawki et al., 2018; Dong et al., 2019), Asian sulfate aerosols significantly weaken the South and East Asian summer monsoons and decrease accordingly the summer precipitation over the Indian subcontinent and northern China (Fig. 2a), through reducing the land-sea thermal gradients and changing the cloud-rain autoconversion processes (second indirect aerosol effect).”

L110: I don't think you need to have arid in front of ACA since arid is in the acronym. This pops up in a few other places as well.

Response: Taken.

L122: In the parentheses it might be good to add some text for clarity so it reads ‘(larger than 65% of total precipitation)’ or something like this.

Response: Taken.

L122-123: I think that ‘with’ can be deleted and the sentence can read ‘Accompanying the increase in convective precipitation there is a sizeable enhancement of 7.4% also shown in the extreme precipitation’.

Response: Taken.

L137: At the end of this sentence the Authors could point to Figures 3a and 3b.

Response: Taken.

Figure 3: In panel b the text in the figure above the bars is quite distracting. Is it really necessary there? Or in general do you need these values within the main text?

Response: Thanks for the suggestions. We have deleted the text in the figure above the bars.

L150: I believe that ‘southwestward wind’ should actually be ‘southwesterly winds’ since the winds are coming from the southwest if I am following correctly.

Response: Taken.

L165: Same comment as L110

Response: Taken.

L172: Same comment as L150

Response: Taken.

L177: It might be good to add to the caption of Figure S6 that these results are from the SULx10Asia experiments for consistency in wording. I think that is what is meant by ‘Asian sulfate-induced changes …’ and so the Authors could just add ‘Asian sulfate-induced changes (SULx10Asia) …’ if that is the case.

Response: Yes, we have added SULx10Asia in the descriptions.

L196: Where the Authors point to Fig. S8 could point directly to Fig. S8b to be consistent with pointing to Fig. S3e in the same sentence.

Response: Taken.

L198-201: This sentence could be re-worded for clarity. Specifically, I think that discover could be replaced with investigate.

Response: Taken.

L224: Same comment as L72-73, what about the spring increase in Figure 1d?

Response: Fig. 1d shows a significant increase in precipitation during the winter, followed by the spring.

L226: Where the Authors point to Fig. S10, it might be good to point directly to the panels referring to which in this case I believe should be Figs. S10b and S10d.

Response: Taken.

Concluding remarks

L233: I think ‘region’ should be included after ‘(ACA)’

Response: Taken.

L240: I think this sentence can be re-worded to read ‘…(AWJS) to shift equatorward’

Response: Taken.

L244: I think ‘southwestward’ should be replaced with ‘southwesterly’ . Also, I think there should be a ‘the’ between ‘by moisture’ so it reads as ‘by the moisture budget analysis’

Response: Taken.

L254: Is it obvious to a broad audience? I encourage the Author’ s to maybe re-word this.

Response: Thanks for your suggestions. We have revised the corresponding descriptions.

Methods

L280: It is not so clear to me what is meant by BASE. If this is an acronym, I think it should be spelt out or if it is just how the Authors are referring to the historical simulations with no modifications/sensitivity testing, I think that should be explained more clearly.

Response: Thanks very much. The BASE is the baseline experiment. We have revised the corresponding as “Nine GCMs with one baseline experiment (BASE) and the perturbed experiments are CanESM2, GISS-E2, HadGEM2, HadGEM3-GA4, IPSL-CM5A, MIROC-SPRINTARS, NCAR-CESM1-CAM4, NCAR-CESM1-CAM5 and NorESM1, as shown in Table S3 (Myhre et al., 2017).”

L284: Why were CanESM2 and HadGEM2 left out?

Response: As mentioned by Myhre et al. (2017) and Liu et al., (2018a), CanESM2 and HadGEM2 have not run these sensitivity experiments with regional aerosols e.g., SULx10Asia and BCx10Asia at present. Therefore, these two models are left out for analyzing the effect of regional aerosols (SULx10Asia).

L285-287: Does ‘15 year running’ and ‘100 year running’ mean that these are 15- and 100-year runs? This is a little confusing to me and I wonder if it can be re-worded for clarity. Perhaps something like: In the PDRMIP experiments, a pair of numerical simulations was conducted for the fixed sea surface temperature (fSST; run for 15 years) and the coupled climate simulations (Coupled; run for 100 years).

Response: Thanks very much for the good suggestions. We have revised the corresponding descriptions as suggested by the Reviewer.

L295: I think the sentence should start ‘Each pair of models…’

Response: Taken.

L317: What is the temporal resolution of the data used for the moisture budget analysis? More broadly it might be good to state the temporal resolution of all data considered within the Methods section.

Response: The temporal resolution of all the data including the moisture budget analysis is monthly mean. In the Method section, we have added the corresponding descriptions about the temporal resolution of the data used here. We have revised the description as “All the simulated data had the monthly temporal resolution and was interpolated to the horizontal resolution by bilinear interpolation with $2.5^{\circ} \times 2.5^{\circ}$, and only the last 10 years of fSST experiments and the last 50 years of Coupled experiments are used to analyze.”

References

- Cherian, R., & Quaas, J. Trends in AOD, clouds, and cloud radiative effects in satellite data and CMIP5 and CMIP6 model simulations over aerosol source regions. *Geophys. Res. Lett.* 47, e2020GL087132 (2020).
- Dong, B.-W., Sutton, R., Shaffrey, L., & Harvey, B. Recent decadal weakening of the summer Eurasian westerly jet attributable to anthropogenic aerosol emissions. *Nat. Commun.* 13, 1148 (2022).
- Du, Y., Li, T., Xie, Z. & Zhu, Z. Interannual variability of the Asian subtropical westerly jet in boreal summer and associated with circulation and SST anomalies. *Clim. Dynam.* 46, 2673-2688 (2016).
- Fan, T., Liu, X., Ma, P.-L., Zhang, Q., Li, Z., Jiang, Y., Zhang, F., Zhao, C., Yang, X., Wu, F., & Wang, Y.: Emission or atmospheric processes? An attempt to attribute the source of large bias of aerosols in eastern China simulated by global climate models, *Atmos. Chem. Phys.* 18, 1395-1417 (2018).
- He, C., Zhou, W., Li, T., Zhou, T., Wang, Y. East Asian summer monsoon enhanced by COVID-19. *Clim Dynam.* doi:10.1007/s00382-022-06247-8 (2022).
- Kripalani, R. H., Oh, J. H., Kang, J. H., Sabade, S. S., Kulkarni, A. Extreme monsoons over East Asia: Possible role of Indian Ocean Zonal Mode. *Theor. Appl. Climatol.* 82, 81-94 (2005).
- Kripalani, R., Ha, K.-J., Ho, C.-H., Oh, J.-H., Preethi, B., Mujumdar, M., Prabhu, A. Erratic Asian Summer Monsoon 2020: COVID-19 Lockdown Initiatives Possible Cause for These Episodes? *Clim Dynam.* 59, 1339-1352 (2022).

- Lamarque, J.-F., et al. The Atmospheric Chemistry and Climate Model Intercomparison Project (ACCMIP): overview and description of models, simulations and climate diagnostics. *Geosci. Model Dev.* 6, 179-206 (2013).
- Li, X., Liu, Y., Wang, M., Jiang, Y., & Dong, X. Assessment of the Coupled Model Intercomparison Project phase 6 (CMIP6) Model performance in simulating the spatial-temporal variation of aerosol optical depth over Eastern Central China. *Atmos. Res.* 261, 105747 (2021).
- Pan, X., Chin, M., Gautam, R., Bian, H., Kim, D., Colarco, P. R., Diehl, T. L., Takemura, T., Pozzoli, L., Tsigaridis, K., Bauer, S., & Bellouin, N. A multi-model evaluation of aerosols over South Asia: common problems and possible causes. *Atmos. Chem. Phys.* 15, 5903-5928 (2015).
- Peng, D., & Zhou, T. Why was the arid and semiarid northwest China getting wetter in the recent decades? *J. Geophys. Res. Atmos.* 122, 9060-9075 (2017).
- Peng, D., Zhou, T., Zhang, L., & Wu, B. Human contribution to the increasing summer precipitation in Central Asia from 1961 to 2013. *J. Clim.* 31(19), 8005-8021 (2018).
- Ramachandran, S., Rupakheti, M., & Cherian, R. Insights into recent aerosol trends over Asia from observations and CMIP6 simulations. *Sci. Total Environ.* 807, 150756 (2022).
- Rodwell, M. J., & Hoskins, B. J. Monsoons and the dynamics of deserts. *Q. J. R. Meteor. Soc.* 122, 1385-1404 (1996).
- Shindell, D. T., et al., Radiative forcing in the ACCMIP historical and future climate simulations. *Atmos. Chem. Phys.* 13, 2939-2974 (2013).
- Wei, W., Zhang, R., Wen, M., & Yang, S. Relationship between the Asian westerly jet stream and summer rainfall over central Asia and north China: Roles of the Indian monsoon and the south Asian high. *J. Clim.* 30(2), 537-552 (2017).
- Zhao, Y., Huang, A., Zhou, Y., Huang, D. Q., Yang, Q., Ma, Y. F., Li, M., & Wei, G. Impact of the middle and upper tropospheric cooling over central Asia on the summer rainfall in the Tarim Basin, China. *J. Clim.* 27(12), 4721-4732 (2014a).
- Zhao, Y., Wang, M., Huang, A., Li, H., Huo, W., & Yang, Q. Relationships between the West Asian subtropical westerly jet and summer precipitation in northern Xinjiang. *Theor. Appl. Climatol.* 116, 403-411 (2014b).

21st Sep 22

Dear Dr Xie,

Your manuscript titled "Teleconnection-like effect on increase of summer precipitation over arid Central Asia by anthropogenic aerosols in heavily polluted regions" has now been seen by 3 reviewers, and I include their comments at the end of this message. They find your work of interest, but some important points are raised. Particularly, you should consider caveating and toning down some of your claims as the potential influence of black carbon aerosol is not accounted for in the PDRMIP simulations (which are dominated by sulfate aerosol) used in your study, and/or include another analysis to account for BC contribution.

We are interested in the possibility of publishing your study in Communications Earth & Environment, but would like to consider your responses to these concerns and assess a revised manuscript before we make a final decision on publication.

We therefore invite you to revise and resubmit your manuscript, along with a point-by-point response that takes into account the points raised. Please highlight all changes in the manuscript text file.

Please use the following link to submit your revised manuscript, point-by-point response to the referees' comments (which should be in a separate document to any cover letter) and the completed checklist:

[link redacted]

We hope to receive your revised paper within six weeks; please let us know if you aren't able to submit it within this time so that we can discuss how best to proceed. If we don't hear from you, and the revision process takes significantly longer, we may close your file. In this event, we will still be happy to reconsider your paper at a later date, as long as nothing similar has been accepted for publication at Communications Earth & Environment or published elsewhere in the meantime.

We understand that due to the current global situation, the time required for revision may be longer than usual. We would appreciate it if you could keep us informed about an estimated timescale for resubmission, to facilitate our planning. Of course, if you are unable to estimate, we are happy to accommodate necessary extensions nevertheless.

Please do not hesitate to contact me if you have any questions or would like to discuss these revisions further. We look forward to seeing the revised manuscript and thank you for the opportunity to review your work.

Best regards,

Akintomide Akinsanola, PhD
Editorial Board Member
Communications Earth & Environment

Clare Davis, PhD
Senior Editor
Communications Earth & Environment

EDITORIAL POLICIES AND FORMATTING

Editorial Policy: [Policy requirements](https://www.nature.com/documents/nr-editorial-policy-checklist.pdf) (Download the link to your computer as a PDF.)

Furthermore, please align your manuscript with our format requirements, which are summarized on the following checklist:

[Communications Earth & Environment formatting checklist](https://www.nature.com/documents/commsj-phys-style-formatting-checklist-article.pdf)

and also in our style and formatting guide [Communications Earth & Environment formatting guide](https://www.nature.com/documents/commsj-phys-style-formatting-guide-accept.pdf) .

***** DATA:** Communications Earth & Environment endorses the principles of the Enabling FAIR data project (<http://www.copdess.org/enabling-fair-data-project/>). We ask authors to make the data that support their conclusions available in permanent, publically accessible data repositories. (Please contact the editor if you are unable to make your data available).

All Communications Earth & Environment manuscripts must include a section titled "Data Availability" at the end of the Methods section or main text (if no Methods). More information on this policy, is available at <http://www.nature.com/authors/policies/data/data-availability-statements-data-citations.pdf>.

DATA SOURCES: All new data associated with the paper should be placed in a persistent repository where they can be freely and enduringly accessed. We recommend submitting the data to discipline-

specific, community-recognized repositories, where possible and a list of recommended repositories is provided at <http://www.nature.com/sdata/policies/repositories>.

If a community resource is unavailable, data can be submitted to generalist repositories such as [figshare](https://figshare.com/) or [Dryad Digital Repository](http://datadryad.org/). Please provide a unique identifier for the data (for example a DOI or a permanent URL) in the data availability statement, if possible. If the repository does not provide identifiers, we encourage authors to supply the search terms that will return the data. For data that have been obtained from publically available sources, please provide a URL and the specific data product name in the data availability statement. Data with a DOI should be further cited in the methods reference section.

REVIEWER COMMENTS:

Reviewer #1 (Remarks to the Author):

Comments

Throughout the paper (including the title and abstract), it is stated that “anthropogenic aerosols...dominate the increase of ACA summer precipitation”. This is not very precise, as it is sulfate aerosol that drives the signal in PDRMIP simulations. Black carbon—another important anthropogenic aerosol species, which is also emitted in large quantities in China and India—actually drives an opposite signal. This needs to be clarified and discussed. For example, real world aerosol emissions include both species. PDRMIP shows BC would act to mute the sulfate signal. But this tug-and-war effect cannot be quantified from PDRMIP simulations. It would appear the authors are implicitly assuming BC is not important (or that sulfate is more important)? It’s unclear what this assumption is based on, outside of the fact sulfate drives a climate signal qualitatively similar to observations.

The authors spend considerable efforts to show CMIP5/6 models underestimate AOD in China/India (or maybe just China). And then use this to explain the weaker ACA signal in CMIP5/6. But again, there are many caveats. If CMIP5/6 underestimate AOD, this could also be due to underestimation of BC. If this dominates the AOD bias, this actually implies CMIP5/6 models overestimate the ACA wetting and southward Jet shift. I understand sulfate contributes more to total AOD than does BC. But my point is there are several caveats and complexities that are not fully acknowledged or discussed.

In a nut shell, this analysis shows that massive increases in sulfate from India and China lead to more ACA precipitation, associated with an equatorward Jet shift. Observations (from ~1960-2005) show a qualitatively similar signal. This implies the possible importance of India/China sulfate. But there are many caveats...

I note a similar study, that you may want to cite, that was recently performed to show how massive increases in sulfate/BC (from PDRMIP) impact precipitation along the West Coast of the US (e.g., California). Interestingly, sulfate versus BC also drove opposite precipitation responses. And the precipitation responses, in turn, were related to jet shifts and altered moisture fluxes. However, this study lacks the strong late 20th century precipitation signal in observations, as found over the ACA (i.e., West Coast precipitation is dominated by internal climate variability).

Allen, R. J., Lamarque, J.-F., Watson-Parris, D., & Olivie, D. (2020). Assessing California wintertime precipitation responses to various climate drivers. *Journal of Geophysical Research: Atmospheres*, 125, e2019JD031736. <https://doi.org/10.1029/2019JD031736>

One needs to be careful with saying the fast response (atmospheric heating/cooling) dominates the ACA signals. The fSST runs do not fix land surface temperatures. The bulk of aerosols are located over the land. So there will be substantial land cooling, which can then lead to atmospheric cooling through altered surface heat fluxes. Sulfate, as opposed to BC, has less direct impact on atmosphere heating/cooling.

Maybe I missed this, but how is CAM5's ERF decomposed into direct and indirect components in Figure S9?

L212. "This unique spatial pattern of ERF is responsible for the insignificant changes in meridional temperature gradients in Fig. S10 and AWJS position in Fig. S3f." How can this be stated definitively? And how do we know this is a "unique" ERF pattern? By comparison with S7b (which is just the MMM ERF)? One reason why the MMM ERF is not negative farther out into the Pacific is because of the experimental design in most PDRMIP models (except CAM5). Emissions versus concentrations. This is yet another caveat...in the real world, emissions from Asia would impact the Pacific (as in CAM5). But this is not allowed in most PDRMIP models, since concentrations (in the Asia box region) are changed. So this is another example of the artificial/idealized construct of PDRMIP simulations. And here, it appears the authors are arguing that the more realistic ERF pattern (which presumably occurs when emissions as opposed to concentrations are perturbed) does not result in a large Jet shift. It's interesting that CAM5 still yields relatively large ACA wetting; any thoughts as to why this is the case?

L235 "Note that black carbon yields a smaller change in AWJS position and precipitation, mainly due to its much lower burden relative to sulfate aerosols". Is burden the most important factor here? There are several...

L286 "Together with our study, these results of sensitivity experiments suggest the need to quantify the relative contributions of the different factors (e.g., anthropogenic aerosols, greenhouse gases, and internal forcings) to the ACA wetting." Isn't this exactly what this study (and the others) have done?

You may want to cite the new PDRMIP data paper:

Myhre, G., Samset, B., Forster, P.M. et al. Scientific data from precipitation driver response model intercomparison project. *Sci Data* 9, 123 (2022). <https://doi.org/10.1038/s41597-022-01194-9>

Reviewer #2 (Remarks to the Author):

My concerns are well addressed by the authors in the revision. I think the manuscript can be accepted for publication.

Reviewer #3 (Remarks to the Author):

I thank the authors for addressing my comments and I support publication of the manuscript.

I have one comment of a minor change I'd like to recommend:

L204: I think it should read 'a local significant decrease'

Response to Reviewer #1:

(1), Throughout the paper (including the title and abstract), it is stated that “anthropogenic aerosols... dominate the increase of ACA summer precipitation”. This is not very precise, as it is sulfate aerosol that drives the signal in PDRMIP simulations. Black carbon—another important anthropogenic aerosol species, which is also emitted in large quantities in China and India—actually drives an opposite signal. This needs to be clarified and discussed. For example, real world aerosol emissions include both species. PDRMIP shows BC would act to mute the sulfate signal. But this tug-and-war effect cannot be quantified from PDRMIP simulations. It would appear the authors are implicitly assuming BC is not important (or that sulfate is more important)? It’s unclear what this assumption is based on, outside of the fact sulfate drives a climate signal qualitatively similar to observations.

Response: Thank the Reviewer very much for the constructive comments and suggestions to further improve our manuscript. The suggestion about the clarified sulfate and black carbon helps us to express our point more clearly. Firstly, we have added sulfate aerosol in the abstract, which increases the ACA summer precipitation and causes an equatorward shift of the summer Asian Westerly Jet Stream (AWJS) through fast responses due to cooling the local atmosphere in mid-latitudes. Additionally, we also added the descriptions about effects of absorbing BC aerosol which can drive an opposite signal in AWJS and local precipitation, in turn can mask the climatic impacts of sulfate. Here, we can separate the impacts of BC and sulfate and show their opposite effect on ACA climate in the PDRMIP runs. However, due to a gap between the idealized 10-fold increase and observational change in aerosols, the PDRMIP cannot quantify the relative contribution of sulfate and BC aerosols in the real world.

Therefore, we revised the Abstract as “Observational evidence has revealed a pronounced precipitation increase during recent decades over the arid Central Asia region (ACA), challenging the paradigm of “dry gets drier, wet gets wetter” under global warming. However, the ACA wetting drivers remain highly controversial, especially for summer. Here, we show, through the analysis of multi-model dataset from the Precipitation Driver and Response Model Intercomparison Project (PDRMIP), that anthropogenic sulfate aerosol over the remote Asian polluted regions significantly increases the ACA summer precipitation, especially convective and extreme precipitation, likely contributing to recent wetting. Mechanistically, Asian sulfate aerosol causes an equatorward shift of the Asian Westerly Jet Stream (AWJS) through fast responses due to cooling the local atmosphere in mid-latitudes, favoring the moisture supply from low-latitudes and moisture flux convergence over ACA, confirmed also by the moisture budget analysis. Contrary to sulfate aerosol, absorbing black carbon drives an opposite signal in AWJS

and local precipitation, which can mask the climatic impacts of sulfate. This teleconnection between ACA precipitation and anthropogenic aerosols in remote Asian polluted regions highlights long-range impacts of anthropogenic aerosols on atmospheric circulations and the hydrological cycle.”

In the Concluding Remarks, we have also added the corresponding description about this tug of war effect between sulfate and BC aerosols as “In contrast to sulfate aerosol, absorbing BC aerosol induces a decrease in ACA precipitation and an AWJS poleward shift, which can mute the climatic effect of sulfate.” Furthermore, we have also discussed this tug of war effect between sulfate and BC aerosols “Additionally, the sulfate and BC aerosols have the opposite effects on the regional precipitation over ACA, which is also evident over the West Coast of the United States (Allen et al., 2020). Combined with influence of a large internal variability of ACA precipitation associated with the El Niño Southern Oscillation and North Atlantic Oscillation (Huang et al., 2013 and Liu et al., 2018b), the tug of war effect between sulfate and BC aerosols makes the detection of specific aerosol climatic effect in observations difficult over this region.”

(2), The authors spend considerable efforts to show CMIP5/6 models underestimate AOD in China/India (or maybe just China). And then use this to explain the weaker ACA signal in CMIP5/6. But again, there are many caveats. If CMIP5/6 underestimate AOD, this could also be due to underestimation of BC. If this dominates the AOD bias, this actually implies CMIP5/6 models overestimate the ACA wetting and southward Jet shift. I understand sulfate contributes more to total AOD than does BC. But my point is there are several caveats and complexities that are not fully acknowledged or discussed.

Response: Yes, we agree with the Review’s point. It is accurate for underestimation of both sulfate and BC aerosols in CMIP5/6 over Asia, especially over the eastern China (Cherian and Quaas, 2020; Li et al., 2021; Ramachandran et al., 2022). We have added the corresponding discussions about the internal variability and the tug of war effect between sulfate and BC aerosols as “Additionally, the sulfate and BC aerosols have the opposite effects on the regional precipitation over ACA, which is also evident over the West Coast of the United States (Allen et al., 2020). Combined with influence of a large internal variability of ACA precipitation associated with the El Niño Southern Oscillation and North Atlantic Oscillation (Huang et al., 2013 and Liu et al., 2018b), the tug of war effect between sulfate and BC aerosols makes the detection of specific aerosol climatic effect in observations difficult over this region.”

(3), In a nut shell, this analysis shows that massive increases in sulfate from India and China lead to more ACA precipitation, associated with an equatorward Jet shift. Observations (from ~1960-2005) show a

qualitatively similar signal. This implies the possible importance of India/China sulfate. But there are many caveats...

Response: Yes, we agree with the Review's point. To accommodate this concern, we have added some discussions about the internal variability and the tug of war effect between sulfate and BC aerosols in Section Concluding Remarks: "Additionally, the sulfate and BC aerosols have the opposite effects on the regional precipitation over ACA, which is also evident over the West Coast of the United States (Allen et al., 2020). Combined with influence of a large internal variability of ACA precipitation associated with the El Niño Southern Oscillation and North Atlantic Oscillation (Huang et al., 2013 and Liu et al., 2018b), the tug of war effect between sulfate and BC aerosols makes the detection of specific aerosol climatic effect in observations difficult over this region."

(4), I note a similar study, that you may want to cite, that was recently performed to show how massive increases in sulfate/BC (from PDRMIP) impact precipitation along the West Coast of the US (e.g., California). Interestingly, sulfate versus BC also drove opposite precipitation responses. And the precipitation responses, in turn, were related to jet shifts and altered moisture fluxes. However, this study lacks the strong late 20th century precipitation signal in observations, as found over the ACA (i.e., West Coast precipitation is dominated by internal climate variability).

Allen, R. J., Lamarque, J.-F., Watson-Parris, D., & Olivie, D. (2020). Assessing California wintertime precipitation responses to various climate drivers. *Journal of Geophysical Research: Atmospheres*, 125, e2019JD031736. <https://doi.org/10.1029/2019JD031736>

Response: Thanks for the useful reference. We have cited the paper and added some discussions about the internal variability and the tug of war effect between sulfate and BC aerosols in Section Concluding Remarks: "Additionally, the sulfate and BC aerosols have the opposite effects on the regional precipitation over ACA, which is also evident over the West Coast of the United States (Allen et al., 2020). Combined with influence of a large internal variability of ACA precipitation associated with the El Niño Southern Oscillation and North Atlantic Oscillation (Huang et al., 2013 and Liu et al., 2018b), the tug of war effect between sulfate and BC aerosols makes the detection of specific aerosol climatic effect in observations difficult over this region."

(5), One needs to be careful with saying the fast response (atmospheric heating/cooling) dominates the ACA signals. The fSST runs do not fix land surface temperatures. The bulk of aerosols are located over the land. So there will be substantial land cooling, which can then lead to atmospheric cooling through

altered surface heat fluxes. Sulfate, as opposed to BC, has less direct impact on atmosphere heating/cooling.

Response: Thanks for pointing this out. We absolutely agree with the Reviewer's point. Sulfate induces atmospheric cooling through reducing the land surface temperature and the surface heat fluxes, whereas BC directly absorbs shortwave radiation to warm the atmosphere. In the manuscript, we have added the corresponding descriptions about these different mechanisms of sulfate and BC.

(6), Maybe I missed this, but how is CAM5's ERF decomposed into direct and indirect components in Figure S9?

Response: We have added the description and references about decomposing direct ERF and indirect ERF in Section Methods: "Note that the effective radiative forcing (ERF) is defined as the total radiative flux differences at the top of atmosphere (TOA) between the BASE and perturbed fSST experiments, which permits rapid adjustments such as atmospheric cooling or heating (Myhre et al., 2013). Direct ERF is simply calculated as the clear-sky radiative flux differences at TOA between these two fSST experiments, whereas indirect ERF is the cloud forcing differences at TOA."

(7), L212. "This unique spatial pattern of ERF is responsible for the insignificant changes in meridional temperature gradients in Fig. S10 and AWJS position in Fig. S3f." How can this be stated definitively? And how do we know this is a "unique" ERF pattern? By comparison with S7b (which is just the MMM ERF)? One reason why the MMM ERF is not negative farther out into the Pacific is because of the experimental design in most PDRMIP models (except CAM5). Emissions versus concentrations. This is yet another caveat...in the real world, emissions from Asia would impact the Pacific (as in CAM5). But this is not allowed in most PDRMIP models, since concentrations (in the Asia box region) are changed. So this is another example of the artificial/idealized construct of PDRMIP simulations. And here, it appears the authors are arguing that the more realistic ERF pattern (which presumably occurs when emissions as opposed to concentrations are perturbed) does not result in a large Jet shift. It's interesting that CAM5 still yields relatively large ACA wetting; any thoughts as to why this is the case?

Response: I agree with the Reviewer's point about different experimental designs in PDRMIP (emissions versus concentrations). We have added the corresponding description and references in Section Results: "The CESM-CAM5 model is emission-driven, whereas the fixed aerosol concentrations are used in most PDRMIP models. The aerosol emissions from the Asia region will affect the downwind area in CESM-CAM5, which leads to a larger indirect ERF over the adjacent oceanic regions. It should be mentioned

that this model significantly overestimates the indirect ERF due to the physical parameterizations of aerosol–cloud interactions e.g., diagnostic precipitation (Gettelman et al., 2013) and dispersion effect on cloud droplet effective radius and cloud-rain autoconversion process (Liu et al., 2002; Xie et al., 2017), which may artificially amplify the aerosol influence over the adjacent oceanic regions.”

Another hypothesis is the "monsoon-desert" mechanism, claiming that the weakened ascent over the Asian monsoon region was directly responsible for the weakened descent over the ACA region (Rodwell et al., 1996; Kripalani et al., 2005). This mechanism can also explain the ACA wetting induced by Asian anthropogenic aerosols due to the weakening Asian monsoon, which is also suitable to the CESM-CAM5 result. This discussion about "monsoon-desert" mechanism has been added in Section Results.

(8), L235 “Note that black carbon yields a smaller change in AWJS position and precipitation, mainly due to its much lower burden relative to sulfate aerosols” . Is burden the most important factor here? There are several...

Response: We agree with the Reviewer’s point about several possible reasons. The lower burden of BC relative to sulfate may be one potential reason. In particular, previous studies have shown that the climatic influence is also dependent on different optical parameters and vertical profiles of aerosols. We have added the corresponding description and references as “Note that BC yields a relatively smaller change in AWJS position and precipitation in PDRMIP results, likely due to its much lower burden relative to sulfate aerosol (Lu et al., 2011; Zhang et al., 2012; Liu et al., 2018a). It is noteworthy that the climatic influence is also dependent on different optical parameters and vertical profiles of aerosols (Bond et al., 2013; Samset et al., 2012). Here, we can separate the impacts of BC and sulfate and show their opposite effect on ACA climate in the PDRMIP runs. However, due to a gap between the idealized 10-fold increase and observational change in aerosols, the PDRMIP cannot quantify the relative contribution of sulfate and BC aerosols in the real world.”

(9), L286 “Together with our study, these results of sensitivity experiments suggest the need to quantify the relative contributions of the different factors (e.g., anthropogenic aerosols, greenhouse gases, and internal forcings) to the ACA wetting.” Isn’ t this exactly what this study (and the others) have done?

Response: Sorry, this sentence is misleading. We have deleted the corresponding sentence.

(10) You may want to cite the new PDRMIP data paper: Myhre, G., Samset, B., Forster, P.M. et al. Scientific data from precipitation driver response model intercomparison project. *Sci Data* 9, 123 (2022). <https://doi.org/10.1038/s41597-022-01194-9>

Response: Thank the Reviewer for this suggestion. We have added the newest reference about the PDRMIP data (Myhre et al., 2022).

Response to Reviewer #2:

My concerns are well addressed by the authors in the revision. I think the manuscript can be accepted for publication.

Response: Thank you very much.

Response to Reviewer #3:

I thank the authors for addressing my comments and I support publication of the manuscript. I have one comment of a minor change I'd like to recommend: L204: I think it should read 'a local significant decrease'

Response: Thanks very much and taken.

References

- Allen, R. J., Lamarque, J.-F., Watson-Parris, D., & Olivie, D. Assessing California wintertime precipitation responses to various climate drivers. *J. Geophys. Res.-Atmos.* 125, e2019JD031736 (2020).
- Bond, T. C., et al. Bounding the role of black carbon in the climate system: A scientific assessment. *J. Geophys. Res.-Atmos.* 118, 5380-552 (2013).
- Cherian, R., & Quaas, J. Trends in AOD, clouds, and cloud radiative effects in satellite data and CMIP5 and CMIP6 model simulations over aerosol source regions. *Geophys. Res. Lett.* 47, e2020GL087132 (2020).
- Gettelman, A., Morrison, H., Terai, C. R. & Wood, R. Microphysical process rates and global aerosol–cloud interactions. *Atmos. Chem. Phys.*, 13, 9855-9867 (2013).
- Li, X., Liu, Y., Wang, M., Jiang, Y., & Dong, X. Assessment of the Coupled Model Intercomparison Project phase 6 (CMIP6) Model performance in simulating the spatial-temporal variation of aerosol optical depth over Eastern Central China. *Atmos. Res.* 261, 105747 (2021).
- Liu, Y. & Daum, P. H. Indirect warming effect from dispersion forcing, *Nature*, 419, 580-581 (2002).
- Myhre G, et al. Scientific data from precipitation driver response model intercomparison project. *Sci. Data* 9, 123 (2022).

Ramachandran, S., Rupakheti, M., & Cherian, R. Insights into recent aerosol trends over Asia from observations and CMIP6 simulations. *Sci. Total Environ.* 807, 150756 (2022).

Samset., B. H. et al. Black carbon vertical profiles strongly affect its radiative forcing uncertainty. *Atmos. Chem. Phys.*, 13, 2423–2434 (2013).

Xie, X. N., Zhang, H., Liu, X. D., Peng, Y. R., & Liu, Y. Sensitivity study of cloud parameterizations with relative dispersion in CAM5.1: Impacts on aerosol indirect effects. *Atmos. Chem. Phys.* 17, 5877-5892 (2017).

21st Nov 22

Dear Dr Xie,

Your manuscript titled "Teleconnection-like effect on increase of summer precipitation over arid Central Asia by anthropogenic aerosols in heavily polluted regions" has now been seen by our reviewers, whose comments appear below. In light of their advice I am delighted to say that we are happy, in principle, to publish a suitably revised version in Communications Earth & Environment under the open access CC BY license (Creative Commons Attribution v4.0 International License).

We therefore invite you to revise your paper one last time to comply with our format requirements and to maximise the accessibility and therefore the impact of your work.

Please note that it may still be possible for your paper to be published before the end of 2022, but in order to do this we will need you to address these points as quickly as possible so that we can move forward with your paper.

EDITORIAL REQUESTS:

SUBMISSION INFORMATION:

OPEN ACCESS:

Communications Earth & Environment is a fully open access journal. Articles are made freely accessible on publication under a [CC BY license](http://creativecommons.org/licenses/by/4.0) (Creative Commons Attribution 4.0 International License). This license allows maximum dissemination and re-use of open access materials and is preferred by many research funding bodies.

For further information about article processing charges, open access funding, and advice and support from Nature Research, please visit <https://www.nature.com/commsenv/article-processing-charges>

At acceptance, you will be provided with instructions for completing this CC BY license on behalf of all authors. This grants us the necessary permissions to publish your paper. Additionally, you will be asked to declare that all required third party permissions have been obtained, and to provide billing

information in order to pay the article-processing charge (APC).

[link redacted]

Best regards,

Akintomide Akinsanola, PhD
Editorial Board Member
Communications Earth & Environment

Clare Davis, PhD
Senior Editor
Communications Earth & Environment

www.nature.com/commsenv/
@CommsEarth

REVIEWERS' COMMENTS:

Reviewer #1 (Remarks to the Author):

The authors have addressed my comments. I recommend publication as is.

Response to Reviewer #1:

REVIEWERS' COMMENTS:

Reviewer #1 (Remarks to the Author):

The authors have addressed my comments. I recommend publication as is.

Response: Thank you very much.